# mitoBK$_{Ca}$ is functionally expressed in murine and human breast cancer cells and potentially contributes to metabolic reprogramming

**Helmut Bischof**[1], **Selina Maier**[1,2], **Piotr Koprowski**[3], **Bogusz Kulawiak**[3], **Sandra Burgstaller**[1,4,5], **Joanna Jasińska**[3], **Kristian Serafimov**[6], **Monika Zochowska**[3], **Dominic Gross**[1], **Werner Schroth**[2,7], **Lucas Matt**[1], **David Arturo Juarez Lopez**[8], **Ying Zhang**[1], **Irina Bonzheim**[9], **Florian A Büttner**[2,7], **Falko Fend**[9], **Matthias Schwab**[2,10,11,12,13], **Andreas L Birkenfeld**[8,14,15], **Roland Malli**[5,16,17], **Michael Lämmerhofer**[6], **Piotr Bednarczyk**[18], **Adam Szewczyk**[3], **Robert Lukowski**[1]*

[1]Department of Pharmacology, Toxicology and Clinical Pharmacy, Institute of Pharmacy, University of Tübingen, Tübingen, Germany; [2]Dr Margarete Fischer-Bosch Institute of Clinical Pharmacology, Stuttgart, Germany; [3]Laboratory of Intracellular Ion Channels, Nencki Institute of Experimental Biology, Polish Academy of Sciences, Warsaw, Poland; [4]NMI Natural and Medical Sciences Institute at the University of Tübingen, Reutlingen, Germany; [5]Center for Medical Research, CF Bioimaging, Medical University of Graz, Graz, Austria; [6]Institute of Pharmaceutical Sciences, Pharmaceutical (Bio-)Analysis, University of Tübingen, Tübingen, Germany; [7]University of Tübingen, Tübingen, Germany; [8]Medical Clinic IV, University Hospital Tübingen, Tübingen, Germany; [9]Institute of Pathology and Neuropathology, University Hospital Tübingen, Tübingen, Germany; [10]iFIT Cluster of Excellence (EXC 2180) "Image-guided and Functionally Instructed Tumor Therapies", University of Tübingen, Tübingen, Germany; [11]Department of Clinical Pharmacology, Universityhostpital of Tübingen, Tübingen, Germany; [12]Department of Biochemistry and Pharmacy, University of Tübingen, Tübingen, Germany; [13]German Cancer Consortium (DKTK), German Cancer Research Center, Partner Site Tübingen, Tübingen, Germany; [14]Institute for Diabetes Research and Metabolic Diseases (IDM) of the Helmholtz Center Munich at the Eberhard Karls University Tübingen, University of Tübingen, Tübingen, Germany; [15]German Center for Diabetes Research (DZD), Neuherberg, Germany; [16]Gottfried Schatz Research Center, Molecular Biology and Biochemistry, Medical University of Graz, Graz, Austria; [17]BioTechMed Graz, Graz, Austria; [18]Department of Physics and Biophysics, Warsaw University of Life Sciences (SGGW), Warsaw, Poland

**\*For correspondence:**
robert.lukowski@uni-tuebingen.de

**Competing interest:** The authors declare that no competing interests exist.

**Abstract** Alterations in the function of K$^+$ channels such as the voltage- and Ca$^{2+}$-activated K$^+$ channel of large conductance (BK$_{Ca}$) reportedly promote breast cancer (BC) development and progression. Underlying molecular mechanisms remain, however, elusive. Here, we provide electrophysiological evidence for a BK$_{Ca}$ splice variant localized to the inner mitochondrial membrane of murine and human BC cells (mitoBK$_{Ca}$). Through a combination of genetic knockdown and knockout along with a cell permeable BK$_{Ca}$ channel blocker, we show that mitoBK$_{Ca}$ modulates overall cellular and mitochondrial energy production, and mediates the metabolic rewiring referred to as the 'Warburg effect', thereby promoting BC cell proliferation in the presence and absence of oxygen.

Additionally, we detect mitoBK$_{Ca}$ and BK$_{Ca}$ transcripts in low or high abundance, respectively, in clinical BC specimens. Together, our results emphasize, that targeting mitoBK$_{Ca}$ could represent a treatment strategy for selected BC patients in future.

## eLife assessment

The large-conductance Ca$^{2+}$ activated K$^+$ channel BK$_{Ca}$ has been reported to promote breast cancer progression. The present study presents **convincing** evidence that an intracellular subpopulation of this channel reprograms breast cancer cells towards the Warburg phenotype, one of the metabolic hallmarks of cancer. This **important** finding advances the field of cancer cell metabolism and has potential therapeutic implications.

## Introduction

Cancer represents a complex disease characterized by unconstrained cell proliferation and the spread of malignant cells in the body (*Kalia, 2015*; *Seyfried and Shelton, 2010*). It is one of the leading causes of death worldwide, with millions of new cases diagnosed each year (*Sung et al., 2021*). Globally, the most prevalent form of cancer represents breast cancer (BC; *Sung et al., 2021*). Despite many available anti-cancer treatments which largely depend on the steroid and epidermal growth factor (HER2) receptor status (*Dunnwald et al., 2007*), cancer cells frequently escape from existing therapies due to adaptions (*Wu et al., 2021b*). Therefore, the identification of novel targets and therapeutic strategies that confer benefits for at least a subset of patients, whose cancer displays specific molecular or cellular features, is of utmost relevance.

Important factors that emerged as new cancer targets are ion channels (*Li and Xiong, 2011*). Especially alterations in the expression levels and function of potassium ion (K$^+$) channels are critically related to cancer malignancy and progression (*Li et al., 2022*). One of these channels represents the calcium ion (Ca$^{2+}$)- and voltage-activated K$^+$ channel of large conductance (BK$_{Ca}$, *KCNMA1*; *Mohr et al., 2022*). Canonical BK$_{Ca}$ channels usually localize in the plasma membrane (PM) of cells, and contribute to the regulation of the cytosolic K$^+$ content, the PM potential ($\Delta\Psi_{PM}$), cell cycle, and cell motility, as well as regulatory volume, changes (*Burgstaller et al., 2022a*). Opening of BK$_{Ca}$ channels results in K$^+$ efflux, increasing the electrochemical driving force for Ca$^{2+}$ entry into the cancer cell and affecting pathological cell growth and survival (*Ouadid-Ahidouch and Ahidouch, 2013*). Accordingly, in BC cells (BCCs), an upregulation of BK$_{Ca}$ has been associated with increased malignancy (*Huang and Jan, 2014*; *Mohr et al., 2022*; *Oeggerli et al., 2012*). However, besides their localization in the PM, several K$^+$ channels, including BK$_{Ca}$, have also been identified in the inner mitochondrial membrane (IMM; mitoBK$_{Ca}$), a topic which has been extensively reviewed recently (*Checchetto et al., 2021*; *Kulawiak and Szewczyk, 2022*; *Szabo and Szewczyk, 2023*; *Szewczyk et al., 2009*; *Wrzosek et al., 2020*). MitoBK$_{Ca}$ was first described by Siemen et al. in a human glioma cell line more than 20 years ago *Siemen et al., 1999*. So far, mitoBK$_{Ca}$ has further been found for example in bronchial epithelial cells (*Dabrowska et al., 2022*), neurons (*Douglas et al., 2006*), skeletal muscle cells (*Skalska et al., 2008*), and in cardiac myocytes (*Xu et al., 2002*). In the latter, a splice variant, the BK$_{Ca}$-DEC isoform containing a unique C-terminal exon of 50 amino acids, forms the functional mitoBK$_{Ca}$ channel at the IMM (*Singh et al., 2013*). Little, however, is known about the molecular identity of mitoBK$_{Ca}$ in other cell types, and mitochondrial localization of BK$_{Ca}$ in BCCs has not been demonstrated so far.

Cancer cells show increased energy demands due to their high proliferation rates. Thus, tumor cells compensate for their elevated energy demand by increasing metabolic activities and adapting to nutrient-poor metabolic niches in the tumor microenvironment, allowing them to overcome oxygen (O$_2$)-dependent mitochondrial metabolism (*Eales et al., 2016*; *Gross et al., 2022*; *Jang et al., 2013*; *Nazemi and Rainero, 2020*). This metabolic switch from oxidative phosphorylation to glycolysis, frequently referred to as the 'Warburg effect', describes the phenomenon that cancer cells rather secrete lactate to the extracellular matrix (ECM), instead of utilizing pyruvate to fuel the TCA cycle (*Warburg, 1924*). This lactate secretion towards the ECM was shown to promote multiple microenvironmental cues eventually causing tumor progression (*de la Cruz-López et al., 2019*). Interestingly, extracellular K$^+$ may also impair effector T-cell function, and, thus, the anti-tumor immune response (*Eil et al., 2016*), while, within the cancer cell, functions of several glycolytic enzymes rely on K$^+$

(*Bischof et al., 2021*; *Gohara and Di Cera, 2016*). Further, the presence of mitoBK$_{Ca}$ contributes to the K$^+$ entry into the mitochondrial matrix to interfere with mitochondrial volume changes and the mitochondrial membrane potential ($\Delta\Psi_{mito}$) (*Krabbendam et al., 2018*). Consequently, (mito)BK$_{Ca}$-dependent mechanisms may severely affect energy production pathways and the resulting energy supply of cancer cells.

To elucidate how BK$_{Ca}$ contributes to increasing BCC malignancy (*Oeggerli et al., 2012*), we utilized BK$_{Ca}$ (*KCNMA1*) pro- and deficient human BCC lines including MDA-MB-453 and MCF-7 cells, and primary murine BCCs derived from the mouse mammary tumor polyoma middle T-antigen (MMTV-PyMT)-induced wild type (WT) or BK$_{Ca}$ knock-out (BK-KO, *Kcnma1-KO*) BC model (*Mohr et al., 2022*). In these cells, either isoforms of BK$_{Ca}$ were expressed, or BK$_{Ca}$ was pharmacologically blocked by paxilline or iberiotoxin, two frequently used inhibitors of BK$_{Ca}$, that either penetrate the cell or act exclusively at PM localized channels, respectively (*Candia et al., 1992*; *Zhou and Lingle, 2014*). These approaches were complemented by patch-clamp experiments and by measuring the cellular ion homeostasis and metabolism with fluorescence live-cell imaging, extracellular flux analysis and liquid chromatography-mass spectrometry (LC-MS)-based methods.

Our results emphasize that the presence of BK$_{Ca}$ in the IMM affects mitochondrial bioenergetics, thereby increasing BCC malignancy. This effect was specifically mediated by the DEC isoform of BK$_{Ca}$, that is, mitoBK$_{Ca}$. Functional expression of BK$_{Ca}$-DEC was validated by single-channel patch-clamp recordings of BCC-derived mitoplasts. Importantly, we also identified BK$_{Ca}$-DEC expression in a subset of BC patient biopsies using nanostring-based mRNA expression analysis. Finally, mitoBK$_{Ca}$ crucially contributed to murine and human BCC proliferation and hypoxic resistance, and its activity increased lactate secretion resulting in higher extracellular acidification rates, even in the presence of O$_2$.

Combined, our analyses provide, for the first time, a mechanistic link between functionally relevant mitochondrial BK$_{Ca}$ isoforms in cancer cells and the promotion of the Warburg effect.

## Results

### Functional characterization of BK$_{Ca}$ expression in BCCs

First, we assessed the functional expression of BK$_{Ca}$ in established human BCC lines by performing whole-cell patch-clamp experiments. Primary BCCs obtained from transgenic, BC-bearing MMTV-PyMT WT or BK-KO mice were used as positive or negative controls, respectively (*Mohr et al., 2022*). Patch-clamp experiments revealed, that depolarizing stimuli delivered in 20 mV increments induced K$^+$ outward currents that were larger in MMTV-PyMT WT compared to BK-KO cells (*Figure 1A and B*, and *Figure 1—figure supplement 1A and B*). To validate that these increased currents were elicited by BK$_{Ca}$, we pharmacologically inhibited the channel by using either paxilline or iberiotoxin (*Candia et al., 1992*; *Zhou and Lingle, 2014*). While the current remained unaffected by these treatments in MMTV-PyMT BK-KO cells (*Figure 1B* and *Figure 1—figure supplement 1B*), peak currents (I$_{max}$) were drastically reduced in MMTV-PyMT WT cells (*Figure 1A* and *Figure 1—figure supplement 1A*). Further, we analyzed the plasma membrane potential ($\Delta\Psi_{PM}$) in these cells using the voltage-sensitive fluorescent dye Dibac4(3) (*Adams and Levin, 2012*) as well as current-clamp experiments. The $\Delta\Psi_{PM}$ was more polarized in MMTV-PyMT WT compared to BK-KO cells (*Figure 1C* and *Figure 1—figure supplement 1C*), as expected, due to the presence of hyperpolarizing BK$_{Ca}$ channels in WT cells (*N'Gouemo, 2014*). Current-clamp experiments unveiled a basal $\Delta\Psi_{PM}$ of –46.4±2.5 mV and –32.2±2.1 mV for MMTV-PyMT WT and BK-KO cells, respectively (*Figure 1—figure supplement 1C*). Iberiotoxin treatment equalized the $\Delta\Psi_{PM}$ of the two genotypes (*Figure 1—figure supplement 1C*). In line with these findings, using a recently developed, genetically encoded K$^+$ sensor, NES lc-LysM GEPII 1.0 (*Bischof et al., 2017*), we found reduced cytosolic K$^+$ concentrations ([K$^+$]$_{cyto}$) in MMTV-PyMT WT cells under basal conditions (83.1±5.8 mM for WT ctrl vs. 121.2±7.1 mM for BK-KO ctrl), which increased to the BK-KO cell level in response to iberiotoxin treatment (109.8±10.8 mM for WT +IBTX vs. 114.6±9.8 mM for BK-KO +IBTX) (*Figure 1—figure supplement 1D and E*), explaining the PM depolarization. Subsequently, we recorded current-voltage relationships using the human BCC lines MDA-MB-453 and MCF-7, expressing either high or low levels of BK$_{Ca}$ mRNA transcripts, respectively (*Mohr et al., 2022*). Analysis of the outward currents activated by depolarization demonstrated a paxilline- and iberiotoxin-sensitive current in MDA-MB-453 cells (*Figure 1D* and *Figure 1—figure supplement 1F*), indicative for functional BK$_{Ca}$ channels in their PM, but not in MCF-7 cells (*Figure 1E*

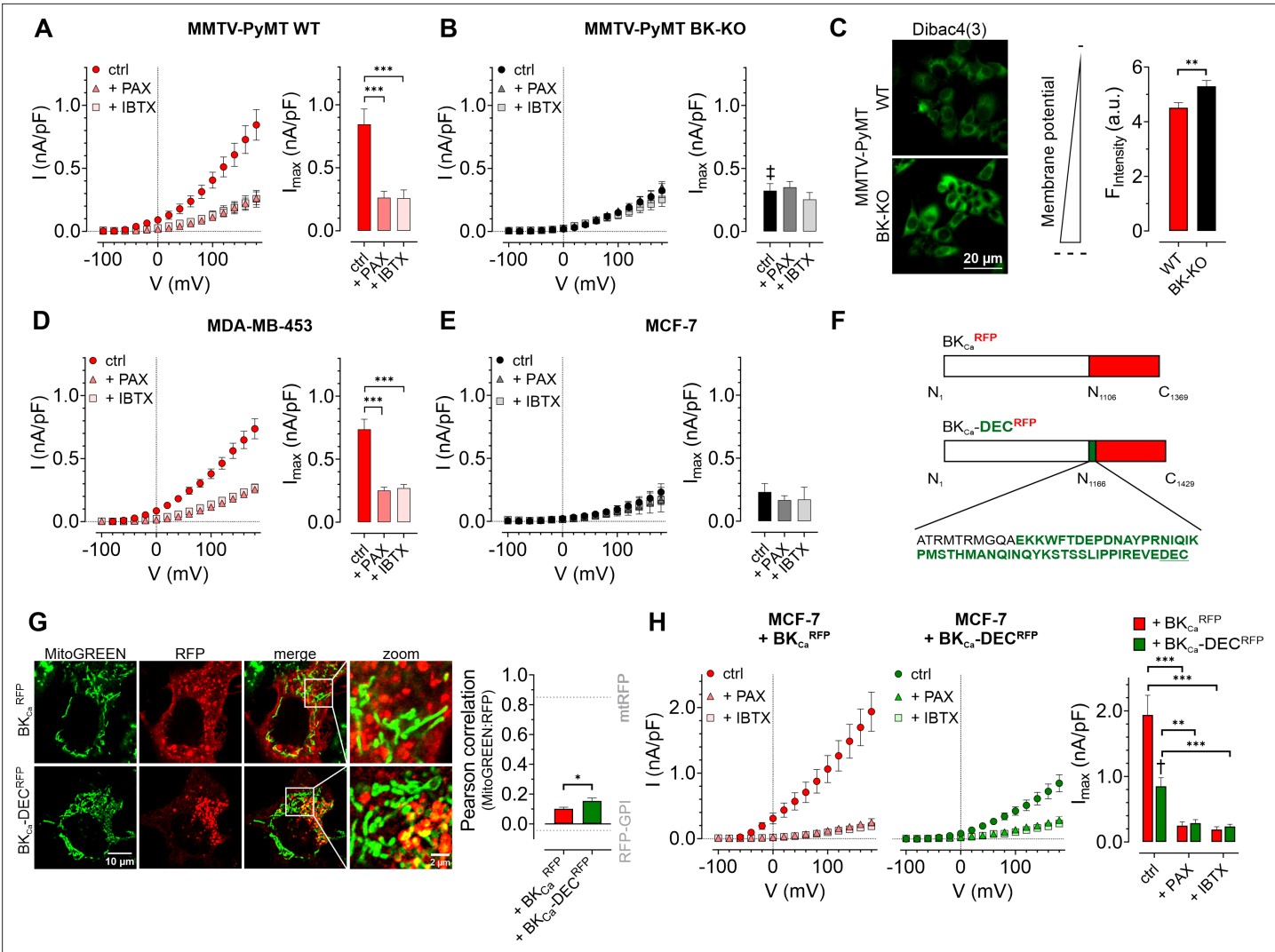

**Figure 1.** Characterization of $BK_{Ca}$ channels in murine and human BCCs. (**A, B**) I-V curves (left) and corresponding maximal currents (right) of MMTV-PyMT WT (**A**) and MMTV-PyMT BK-KO (**B**) cells, either under control conditions, or in the presence of paxilline or iberiotoxin. Data represents average ± SEM. n (cells) = 15 WT ctrl, 17 WT +PAX, 17 WT +IBTX, 16 BK-KO ctrl, 17 BK-KO +PAX, 19 BK-KO +IBTX. ***p≤0.001, Brown-Forsythe and Welch ANOVA test followed by Games-Howell's multiple comparison test. ‡P≤0.001 compared to respective WT condition, Welch's t-test. (**C**) Representative fluorescence images (left) and statistics (right) of MMTV-PyMT WT and BK-KO cells loaded with the $\Delta\Psi_{PM}$ sensitive dye Dibac4(3). N = 6 independent experiments, **p≤0.01, Unpaired t-test. (**D**) I-V curves (left) and maximal currents (right) of MDA-MB-453 cells, either under control conditions, or in the presence of paxilline or iberiotoxin. Data represents average ± SEM. n (cells) = 30 ctrl, 22 +PAX, 24 +IBTX. ***p≤0.001, Kruskal-Wallis test followed by Dunn's multiple comparison test. (**E**) I-V curves (left) and maximal currents (right) of MCF-7 cells, either under control conditions, or in the presence of paxilline or iberiotoxin. Data shows average ± SEM. n (cells) = 16 ctrl, 20 +PAX, 15 +IBTX. (**F**) Schematic representation of constructs used for over-expression in MCF-7 cells. The DEC exon is indicated in green. (**G**) Representative images (left) of MCF-7 cells either expressing $BK_{Ca}^{RFP}$ (upper) or $BK_{Ca}$-$DEC^{RFP}$ (lower), additionally stained with MitoGREEN. Average Pearson correlations ± SEM of MitoGREEN and RFP of $BK_{Ca}$ or $BK_{Ca}$-DEC are shown. n (cells) = 17 $BK_{Ca}$-RFP, 22 $BK_{Ca}$-$DEC^{RFP}$. *p≤0.05, Unpaired t-test. (**H**) I-V curves (left and middle) and corresponding maximal currents (right) of MCF-7 cells expressing $BK_{Ca}^{RFP}$ (left) or $BK_{Ca}$-$DEC^{RFP}$ (middle), respectively, either under control conditions, or in the presence of paxilline or iberiotoxin. Data represents average ± SEM. n (cells) = 18 $BK_{Ca}^{RFP}$ ctrl, 14 $BK_{Ca}^{RFP}$ +PAX, 19 $BK_{Ca}^{RFP}$ +IBTX, 18 $BK_{Ca}$-$DEC^{RFP}$ ctrl, 21 $BK_{Ca}$-$DEC^{RFP}$ +PAX, 18 $BK_{Ca}$-$DEC^{RFP}$ +IBTX. **P≤0.01, ***p≤0.001, Brown-Forsythe and Welch ANOVA test followed by Games-Howell's multiple comparison test. †p≤0.01 between ctrl conditions, Welch's t-test.

The online version of this article includes the following source data and figure supplement(s) for figure 1:

**Source data 1.** Numerical values underlying the data shown in *Figure 1*.

**Figure supplement 1.** Representative whole-cell patch-clamp traces and colocalization analysis in BCCs.

**Figure supplement 1—source data 1.** Numerical values underlying the data shown in *Figure 1*.

and *Figure 1—figure supplement 1G*), which is well in line with previous studies (*Mohr et al., 2022*). Based on the almost non-existent level of $BK_{Ca}$-mediated PM currents in MCF-7 cells, we utilized these cells to express different $BK_{Ca}$ isoforms fused to a red fluorescent protein (RFP). Therefore, we either used an earlier identified mito$BK_{Ca}$ isoform (*Singh et al., 2013*), namely $BK_{Ca}$-DEC$^{RFP}$, or the same channel lacking C-terminal amino acids including the DEC exon, hereinafter referred to as $BK_{Ca}^{RFP}$ (*Figure 1F*). We hypothesized, that, in analogy to cardiac myocytes (*Singh et al., 2013*), the expression of $BK_{Ca}$-DEC$^{RFP}$ in MCF-7 cells may yield a functional channel present in the IMM. To test this, MCF-7 cells were first transiently transfected either with RFP fused to a glycosylphosphatidylinositol (GPI)-anchor (RFP-GPI) directing RFP to the PM, or an RFP linked to a cytochrome c oxidase subunit 8 (COX8) mitochondrial leading sequence, yielding a mitochondrial targeted fusion protein (mtRFP; *Figure 1—figure supplement 1H* and grey dotted lines in *Figure 1G*). Analysis of the colocalization of mtRFP with MitoGREEN, a dye that specifically stains the mitochondrial matrix, resulted in high colocalization scores, while a low overlap was observed for RFP-GPI transfected MCF-7 cells (*Figure 1—figure supplement 1H* and *Figure 1G*, grey dotted lines). Subsequently, the same experiments were performed with MCF-7 cells expressing $BK_{Ca}^{RFP}$ or $BK_{Ca}$-DEC$^{RFP}$. Although the mitochondrial localization score was lower for $BK_{Ca}$-DEC$^{RFP}$ compared to mtRFP (compare *Figure 1G* and *Figure 1—figure supplement 1H*), this $BK_{Ca}$ variant showed a significantly higher overlap with the mitochondrial dye than $BK_{Ca}^{RFP}$ (*Figure 1G* and *Figure 1—figure supplement 1I*). Despite rather moderate colocalization scores of $BK_{Ca}$-DEC$^{RFP}$ with MitoGREEN, on the level of single mitochondria, the RFP signal derived from $BK_{Ca}$-DEC$^{RFP}$ surrounded the MitoGREEN fluorescence signal originating from the mitochondrial matrix, as expected for a $K^+$ channel located in the IMM (*Figure 1G* and *Figure 1—figure supplement 1I*). As not all RFP signal in the $BK_{Ca}$-DEC$^{RFP}$ overexpressing MCF-7 originated from mitochondria, we investigated the PM localization of the two channel isoforms by patch-clamp. Compared to native MCF-7 cells (*Figure 1E*), both, $BK_{Ca}^{RFP}$ and $BK_{Ca}$-DEC$^{RFP}$, increased the PM outward current (*Figure 1H* and *Figure 1—figure supplement 1J and K*). Despite comparable expression levels of the RFP signals (*Figure 1—figure supplement 1L*), presence of $BK_{Ca}^{RFP}$ caused significantly bigger currents across the PM compared to $BK_{Ca}$-DEC$^{RFP}$ (*Figure 1H*), indicative either for (i) higher intracellular abundance or (ii) major functional differences of $BK_{Ca}$-DEC$^{RFP}$. Importantly, the conductance of both channels was sensitive to the $BK_{Ca}$ blockers paxilline and iberiotoxin, as treatment of the cells with both compounds resulted in PM conductance values comparable to those of native MCF-7 cells (*Figure 1E and H* and *Figure 1—figure supplement 1G, J and K*).

## $BK_{Ca}$ modulates global and subcellular $Ca^{2+}$ homeostasis in BCCs

As $BK_{Ca}$ potentially affects cellular $Ca^{2+}$ fluxes (*Ouadid-Ahidouch and Ahidouch, 2013*), we next investigated the cytosolic (*Figure 2A–C*), endoplasmic reticulum (ER) (*Figure 2D–F*), and mitochondrial (*Figure 2G–I*) $Ca^{2+}$ homeostasis in these cells. First, we assessed changes in the cytosolic $Ca^{2+}$ concentration ($[Ca^{2+}]_{cyto}$) over-time in response to cell stimulation with the purinergic receptor agonist adenosine-5′-triphosphate (ATP) (*Müller et al., 2020*) using the fluorescent $Ca^{2+}$ indicator Fura-2 (*Grynkiewicz et al., 1985*). Of note, extracellular ATP, released for example from necrotic cells in the tumor microenvironment, could be of pathophysiological relevance for BCCs (*Di Virgilio et al., 2018*), and all cell lines used in our study were previously reported to respond to ATP stimulation with a significant increase in intracellular $Ca^{2+}$ (*Chadet et al., 2014*; *Gross et al., 2022*; *Klijn et al., 2015*). These experiments were either performed under control conditions (*Figure 2A*) or in the presence of paxilline (*Figure 2—figure supplement 1A*) or iberiotoxin (*Figure 2—figure supplement 1B*). Analysis of the Fura-2 fluorescence emission ratio showed significantly elevated basal (*Figure 2A*) and ATP-elicited maximal $[Ca^{2+}]_{cyto}$ responses (*Figure 2—figure supplement 1C*) in MMTV-PyMT WT compared to BK-KO cells under control conditions. Interestingly, the elevated basal $[Ca^{2+}]_{cyto}$ of WT cells was reduced to the BK-KO cell level in response to paxilline (*Figure 2A* and *Figure 2—figure supplement 1A*), but not by iberiotoxin (*Figure 2A* and *Figure 2—figure supplement 1B*), which is a peptide-based pore-blocking toxin that cannot pass the PM (*Candia et al., 1992*). To validate the observed basal differences in $[Ca^{2+}]_{cyto}$, we additionally performed ionomycin-based calibrations by chelating intracellular $Ca^{2+}$, followed by Fura-2 saturation (*Figure 2—figure supplement 1D - F*). These experiments confirmed the observed differences in basal $[Ca^{2+}]_{cyto}$, with $[Ca^{2+}]_{cyto}$ of 170.2±8.0 nM, 98.40±4.8 nM and 178.5±13.5 nM for MMTV-PyMT WT, and 102.3±4.9 nM, 96.49±3.3 nM and 114.9±4.3 nM for MMTV-PyMT BK-KO cells, under control conditions, or in the presence of paxilline

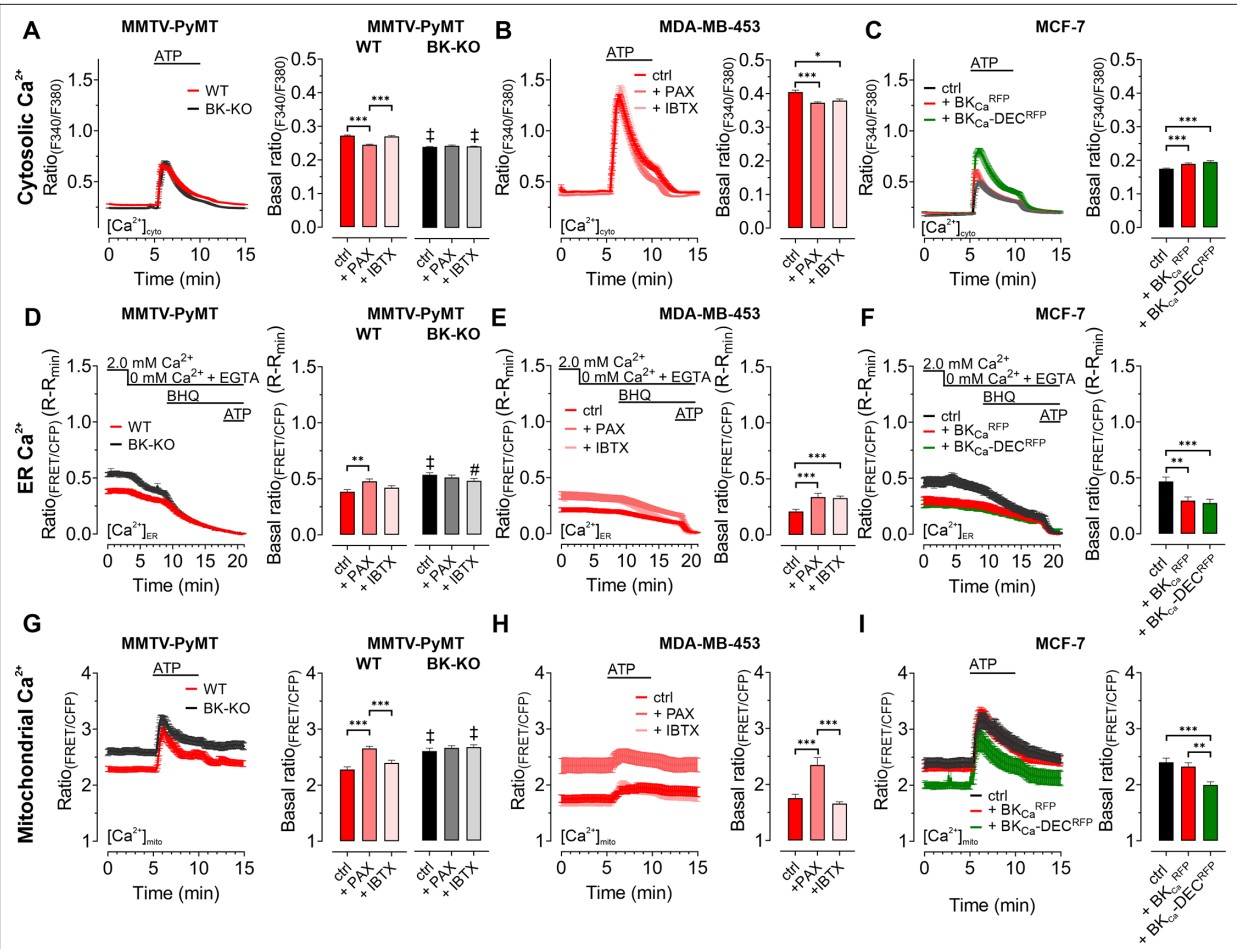

**Figure 2.** BK$_{Ca}$ modulates the subcellular Ca$^{2+}$ homeostasis in BCCs. Cytosolic (**A – C**), endoplasmic reticulum (ER) (**D – F**), and mitochondrial Ca$^{2+}$ dynamics (**G – I**) over-time of MMTV-PyMT WT and MMTV-PyMT BK-KO cells (**A, D, G**), MDA-MB-453 cells (**B, E, H**) or MCF-7 cells (**C, F, I**). All data represent average ± SEM. At time points indicated in the panels, cytosolic and mitochondrial Ca$^{2+}$ alterations were evoked by extracellular stimulation with ATP (**A – C, G – I**), or by Ca$^{2+}$ depletion of the ER using Ca$^{2+}$-free buffer containing the Ca$^{2+}$ chelator EGTA, followed by administration of the SERCA inhibitor BHQ prior to ATP administration (**D – F**). MMTV-PyMT (**A, D, G**) and MDA-MB-453 (**B, E, H**) cells were either measured under control conditions, or in the presence of paxilline (+PAX) or iberiotoxin (+IBTX). MCF-7 cells (**C, F, I**) either expressed an RFP (ctrl), BK$_{Ca}$$^{RFP}$, or BK$_{Ca}$-DEC$^{RFP}$. N (independent experiments) / n (cells analyzed) = (**A**): 17/784 WT ctrl, 18/857 BK-KO ctrl, 6/300 WT +PAX, 6/300 BK-KO +PAX, 5/318 WT +IBTX, 5/304 BK-KO +IBTX, (**B**): 4/151 ctrl, 4/132 +PAX, 4/87 +IBTX, (**C**): 5/111 ctrl, 5/117 +BK$_{Ca}$$^{RFP}$, 5/91 +BK$_{Ca}$-DEC$^{RFP}$, (**D**): 14/116 WT ctrl, 13/117 BK-KO ctrl, 8/71 WT +PAX, 8/92 BK-KO +PAX, 6/102 WT +IBTX, 6/86 BK-KO +IBTX, (**E**): 7/44 ctrl, 9/34 +PAX, 5/49 +IBTX, (**F**): 4/25 ctrl, 4/35 +BK$_{Ca}$$^{RFP}$, 4/38 +BK$_{Ca}$-DEC$^{RFP}$, (**G**): 11/47 WT ctrl, 12/86 BK-KO ctrl, 6/46 WT +PAX, 6/58 BK-KO +PAX, 5/59 WT +IBTX, 4/43 BK-KO +IBTX, (**H**): 8/33 ctrl, 8/28 +PAX, 5/22 +IBTX, (**I**): 5/28 ctrl, 4/27 +BK$_{Ca}$$^{RFP}$, 4/24 +BK$_{Ca}$-DEC$^{RFP}$. **p≤0.01, ***p≤0.001. Kruskal-Wallis test followed by Dunn's MC test (**A, B, C, D, I**), One-Way ANOVA test followed by Tukey's MC test (**E, F, G**) or Brown-Forsythe ANOVA test followed by Games-Howell's MC test (**H**). #p≤0.05, ‡p≤0.001 compared to respective WT condition, Mann-Whitney test (**A, D**) ctrl in (**G**) or Welch's t test (+IBTX in **G**).

The online version of this article includes the following source data and figure supplement(s) for figure 2:

**Source data 1.** Numerical values underlying the data shown in *Figure 2*.

**Figure supplement 1.** Cytosolic-, ER-, and mitochondrial Ca$^{2+}$ homeostasis is altered by functional BK$_{Ca}$ expression in BCCs.

**Figure supplement 1—source data 1.** Numerical values underlying the data shown in *Figure 2*.

or iberiotoxin, respectively. Subsequently, basal and ATP-elicited maximal [Ca$^{2+}$]$_{cyto}$ transients were recorded in the human BCC line MDA-MB-453, where both BK$_{Ca}$ blockers reduced the basal [Ca$^{2+}$]$_{cyto}$ (*Figure 2B*), while maximal [Ca$^{2+}$]$_{cyto}$ was not altered by these treatments (*Figure 2—figure supplement 1G*). Again, ionomycin-based experiments confirmed the BK$_{Ca}$-modulation dependent reduction of [Ca$^{2+}$]$_{cyto}$ (*Figure 2—figure supplement 1H*), with [Ca$^{2+}$]$_{cyto}$ of 110.7±8.0 nM, 64.1±4.8 nM and 78.2±13.5 nM under control, paxilline- or iberiotoxin-treated conditions, respectively. Eventually, [Ca$^{2+}$]$_{cyto}$ was assessed in MCF-7 cells expressing or lacking the different BK$_{Ca}$ isoforms (*Figure 2C*).

These experiments confirmed previous findings with the other BCC lines, as the expression of both splice variants increased the maximal (*Figure 2C* and *Figure 2—figure supplement 1I*) and basal $[Ca^{2+}]_{cyto}$ (*Figure 2—figure supplement 1J*). $[Ca^{2+}]_{cyto}$ in MCF-7 cells was found to be 31.5±3.7 nM, 58.1±6.4 nM and 52,9±5.6 nM in MCF-7 control cells expressing RFP, or MCF-7 cells expressing $BK_{Ca}^{RFP}$ or $BK_{Ca}$-$DEC^{RFP}$, respectively.

Next, we visualized changes in the ER $[Ca^{2+}]$ ($[Ca^{2+}]_{ER}$). Therefore, BCCs were transfected with D1ER, an established genetically encoded, FRET-based $Ca^{2+}$ sensor targeted to the lumen of the ER (*Palmer et al., 2004*). $[Ca^{2+}]_{ER}$ was depleted by extracellular $Ca^{2+}$ removal and chelation by EGTA, followed by inhibition of the sarcoplasmic endoplasmic reticulum $Ca^{2+}$ ATPase (SERCA) with BHQ and activation of inositol 1,4,5-trisphosphate ($IP_3$) receptors upon purinergic receptor stimulation with ATP (*Lape et al., 2008*; *Müller et al., 2020*; *Salter and Hicks, 1995*). Experiments were either performed under control conditions (*Figure 2D–F*), or in the presence of paxilline or iberiotoxin for MMTV-PyMT (*Figure 2D* and *Figure 2—figure supplement 1K, L*)**,** MDA-MB-453 (*Figure 2E*), and MCF-7 cells expressing $BK_{Ca}^{RFP}$ or $BK_{Ca}$-$DEC^{RFP}$ (*Figure 2F*). Throughout all BCCs investigated, expression of $BK_{Ca}$ reduced $[Ca^{2+}]_{ER}$, potentially indicating that the channel was (i) functional in this cell compartment and (ii) involved in regulating the $Ca^{2+}$ homeostasis of the ER. In MMTV-PyMT WT cells, $[Ca^{2+}]_{ER}$ was restored by the cell-permeable $BK_{Ca}$ inhibitor paxilline (*Figure 2D* and *Figure 2—figure supplement 1K*) but not by the cell-impermeable iberiotoxin (*Figure 2D* and *Figure 2—figure supplement 1L*), while both inhibitors restored the $[Ca^{2+}]_{ER}$ in MDA-MB-453 cells (*Figure 2E*). Accordingly, overexpression of both RFP-tagged $BK_{Ca}$ isoforms in MCF-7 cells depleted the $[Ca^{2+}]_{ER}$ (*Figure 2F*).

$[Ca^{2+}]_{cyto}$ and $[Ca^{2+}]_{ER}$ reportedly affect the mitochondrial $[Ca^{2+}]$ ($[Ca^{2+}]_{mito}$) (*Wacquier et al., 2019*). Since an accumulation of $K^+$ within the mitochondrial matrix may oppose mitochondrial $Ca^{2+}$ uptake (*Checchetto et al., 2021*), we assessed the effects of functional $BK_{Ca}$ expression on $[Ca^{2+}]_{mito}$ utilizing 4mtD3cpV, a genetically encoded $Ca^{2+}$ sensor targeted to the mitochondrial matrix (*Palmer et al., 2006*). Again, BCCs were treated with ATP to investigate the $[Ca^{2+}]_{mito}$ upon cell stimulation. In MMTV-PyMT WT (*Figure 2G*) as well as MDA-MB-453 cells (*Figure 2H*), functional expression of $BK_{Ca}$ reduced basal $[Ca^{2+}]_{mito}$. Interestingly, in these two BCC types, basal and maximally elicited $[Ca^{2+}]_{mito}$ peaks increased in response to paxilline (*Figure 2G and H* and *Figure 2—figure supplement 1M—P*), but not iberiotoxin treatment (*Figure 2G and H* and *Figure 2—figure supplement 1M—P*). These findings confirm a role of intracellular located $BK_{Ca}$ channels in modulating $[Ca^{2+}]_{mito}$ dynamics. Importantly, in MCF-7 cells, basal $[Ca^{2+}]_{mito}$ was only affected upon expression of the mitochondrially targeted $BK_{Ca}$-$DEC^{RFP}$, but not $BK_{Ca}^{RFP}$ (*Figure 2I*), whereas neither of the two isoforms affected the maximal $[Ca^{2+}]_{mito}$ (*Figure 2—figure supplement 1Q*). Presumably, by facilitating $K^+$ fluxes across the IMM, $BK_{Ca}$-$DEC^{RFP}$ expression reduces the driving force for $Ca^{2+}$ uptake and thus the resulting $Ca^{2+}$ signals in the mitochondrial matrix. Basal differences in all the utilized cell lines under the different conditions were additionally validated by ionomycin treatment, followed by intracellular $Ca^{2+}$ chelation (*Figure 2—figure supplement 1R-V*).

In total, our $[Ca^{2+}]$ imaging approaches emphasize that different $BK_{Ca}$ isoforms at different localizations, i.e. the PM or intracellular organelles, may either amplify or weaken $[Ca^{2+}]_{cyto}$ and $[Ca^{2+}]_{ER}$ signals, respectively. These opposing effects are expected, as $BK_{Ca}$-mediated uptake of $K^+$ into the ER should limit the $Ca^{2+}$ uptake capacity of this subcellular compartment, while functional channel expression at the PM might increase the driving force for $Ca^{2+}$ influx to fuel $[Ca^{2+}]_{cyto}$. $[Ca^{2+}]_{mito}$, however, seems to be exclusively and effectively suppressed by the activity of intracellularly located $BK_{Ca}$ variants in MMTV-PyMT WT and MDA-MB-453 cells and solely by the expression of $BK_{Ca}$-$DEC^{RFP}$ in MCF-7 cells.

## The metabolic activity of murine and human BCCs is modulated by intracellular $BK_{Ca}$

Given the involvement of $Ca^{2+}$ and $K^+$ in regulating key features of cellular metabolism (*Bischof et al., 2021*; *Dejos et al., 2020*; *Gohara and Di Cera, 2016*), we proposed, that $BK_{Ca}$ may alter metabolic activities of BCCs (*Rossi et al., 2019*). Hence, we analyzed the extracellular acidification rate (ECAR) as a measure of lactate secretion, that is glycolytic activity in MMTV-PyMT and MDA-MB-453 cells (*Burgstaller et al., 2021a*). ECAR measurements unveiled an increased basal ECAR of MMTV-PyMT WT compared to BK-KO cells (*Figure 3A and B*). Subsequently, we performed a mitochondrial stress test by injection of Oligomycin-A, FCCP, and Antimycin-A, which increased ECARs in both cell types due to ATP-synthase inhibition, mitochondrial uncoupling and complex III blockade, respectively

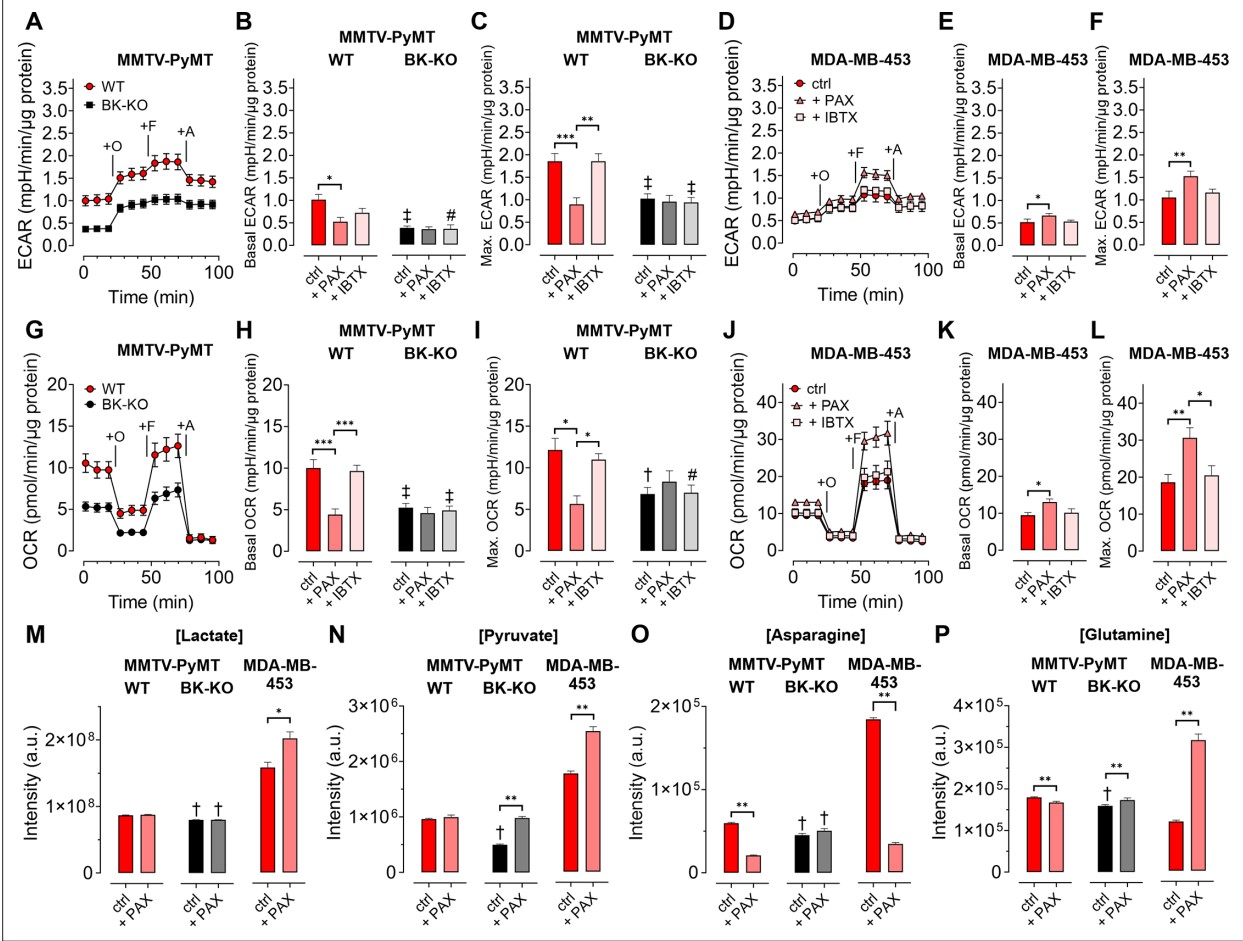

**Figure 3.** BK$_{Ca}$ channels alter the metabolic activity of BCCs. (**A, D**) Average ECAR over-time ± SEM of MMTV-PyMT WT (**A**, red) and BK-KO cells (**A**, black), or MDA-MB-453 cells (**D**) in response to administration of Oligomycin-A (+O), FCCP (+F) and Antimycin-A (+A) at time points indicated. (**B, E**) Average basal and (**C, F**) maximal ECAR ± SEM of MMTV-PyMT WT (**B, C**, left) and BK-KO cells (**B, C**, right), or MDA-MB-453 cells (**E, F**) under control conditions, or in the presence of paxilline or iberiotoxin. (**G, J**) Average OCR over-time ± SEM of MMTV-PyMT WT (**G**, red) and BK-KO cells (**G**, black), or MDA-MB-453 cells (**J**) in response to administration of Oligomycin-A (+O), FCCP (+F) and Antimycin-A (+A) at time points indicated. (**H, K**) Average basal and (**I, L**) maximal OCR ± SEM of MMTV-PyMT WT (**H, I**, left) and BK-KO cells (**H, I**, right), or MDA-MB-453 cells (**K, L**) under control conditions, or in the presence of paxilline or iberiotoxin. (**M – P**) LC-MS-based determination of major glycolytic and mitochondrial metabolites in particular Lactate [Lactate]; (**M**), Pyruvate [Pyruvate]; (**N**), Asparagine [Asparagine]; (**O**), and Glutamine [Glutamine]; (**P**), of MMTV-PyMT WT (left panels), BK-KO (middle panels) and MDA-MB-453 cells (right panels), either under control conditions or after cell cultivation with paxilline. N (independent experiments) = (**A, B, C, G, H, I**): 7 WT ctrl and BK-KO, 3 for all others, (**D, E, F, J, K, L**): 3 for all, (**M – P**): 7 for BK-KO ctrl, 6 for all others. *p≤0.05, **p≤0.01, ***p≤0.001, Kruskal-Wallis test followed by Dunn's MC test (**B, E, F, I**), Brown-Forsythe and Welch ANOVA test followed by Games-Howell's MC test (**C, H**), One-Way ANOVA test followed by Tukey's MC test (**K, L**) or Mann-Whitney test (**M – P**). #p≤0.05, †p≤0.01, ‡p≤0.001, to respective WT condition, Mann-Whitney test (**B, C**) +IBTX in (**I**), (**M – P**), Welch's t-test (ctrl in **H**) and (**I**) or Unpaired t-test (+IBTX in **H**).

The online version of this article includes the following source data and figure supplement(s) for figure 3:

**Source data 1.** Numerical values underlying the data shown in *Figure 3*.

**Figure supplement 1.** Paxilline and iberiotoxin differentially modulate ECAR and OCR in MMTV-PyMT WT and BK-KO cells.

**Figure supplement 1—source data 1.** Numerical values underlying the data shown in *Figure 3*.

(*Figure 3A*). Maximal ECAR, received upon FCCP injection, was elevated in WT compared to BK-KO cells (*Figure 3C*). Besides this evidence derived from a gene-targeted BK$_{Ca}$ channel-deficient model, we used paxilline or iberiotoxin as pharmacological BK$_{Ca}$ modulators (*Figure 3—figure supplement 1A and B*). Interestingly, only paxilline, but not iberiotoxin treatment equalized basal (*Figure 3B*) and maximal ECARs (*Figure 3C*) between WT and BK-KO, suggesting that intracellular BK$_{Ca}$ channels stimulate the glycolytic activity of BCCs. Moreover, in MDA-MB-453 cells iberiotoxin treatment did neither affect basal (*Figure 3D and E*) nor maximal ECARs (*Figure 3F*). Contrary to MMTV-PyMT WT cells,

however, paxilline increased basal and maximal ECAR in these cells (*Figure 3E and F*), suggesting that the cell lines examined strongly differ in their metabolic settings or in other $H^+$ releasing processes contributing to extracellular acidification.

Next, we investigated the oxygen consumption rates (OCRs; *Burgstaller et al., 2021a*). (*Figure 3G*). Interestingly, the increased basal and maximal ECAR (*Figure 3A–C*) in MMTV-PyMT WT cells correlated with a higher basal (*Figure 3H*) and maximal (*Figure 3I*) OCR compared to BCCs lacking $BK_{Ca}$. Paxilline treatment reduced the basal and maximal OCR of $BK_{Ca}$ proficient MMTV-PyMT cells to the BK-KO level (*Figure 3H and I* and *Figure 3—figure supplement 1C*), while iberiotoxin did not have any effect (*Figure 3H and I* and *Figure 3—figure supplement 1D*) again suggesting that that the latter toxin cannot reach the metabolism-relevant population of intracellular $BK_{Ca}$ channels. Again, in MDA-MB-453 cells, paxilline treatment had an opposite effect to the observations that were made in MMTV-PyMT WT cells, as paxilline treatment increased the OCR (*Figure 3J–L*), potentially due to the increased supply of oxidative phosphorylation with glycolytic substrates (*Figure 3D–F*). More importantly, regardless of this difference between the BCC lines iberiotoxin treatment did consistently not affect the OCR of MDA-MB-453 cells (*Figure 3J–L*).

To confirm that $BK_{Ca}$ regulates the bioenergetic profile of BCCs, we subsequently applied LC-MS-based metabolomics according to the workflow presented in *Figure 3—figure supplement 1E*. We included typical analytes of glycolysis such as lactate (*Figure 3M*) and pyruvate (*Figure 3N*), as well as selected metabolites of mitochondrial metabolism including asparagine (*Figure 3O*) and glutamine (*Figure 3P*). Single time-point measurements confirmed the metabolic differences between MMTV-PyMT WT and BK-KO cells and further demonstrated that $BK_{Ca}$ inhibition by paxilline directly affected the concentrations of these metabolites (*Figure 3M–P*). As observed before by the ECAR and OCR measurements, MMTV-PyMT and MDA-MB-453 cells responded, however, differently to paxilline treatment (*Figure 3M–P*). In summary, our data strengthen the notion that intracellular $BK_{Ca}$ modulates the cellular energy balance of murine and human BCCs.

## $BK_{Ca}$ alters the mitochondrial function of BCCs

To clarify how $BK_{Ca}$ regulates BCC cell metabolic activities, we examined cellular bioenergetics in real time using single-cell fluorescence microscopy. First, we assessed the mitochondrial membrane potential ($\Delta\Psi_{mito}$) of MMTV-PyMT and MDA-MB-453 cells using TMRM under control and $BK_{Ca}$ inhibitor-treated conditions, followed by depolarization of $\Delta\Psi_{mito}$ using the proton ionophore FCCP (*Joshi and Bakowska, 2011*). These measurements unveiled a less polarized $\Delta\Psi_{mito}$ of MMTV-PyMT WT cells compared to BK-KO cells under control conditions (*Figure 4A and B*). Paxilline, but not iberiotoxin, equalized $\Delta\Psi_{mito}$ between MMTV-PyMT WT and BK-KO cells (*Figure 4B*). Identical results were obtained in MDA-MB-453 cells (*Figure 4C and D*).

Interestingly, we found that $\Delta\Psi_{mito}$ of MMTV-PyMT cells was tightly dependent on the extracellular glucose concentration ($[GLU]_{ex}$) (*Figure 4—figure supplement 1A*). Under high $[GLU]_{ex}$ conditions (25.0 mM), the difference in $\Delta\Psi_{mito}$ between MMTV-PyMT WT and BK-KO cells disappeared, as $\Delta\Psi_{mito}$ increased significantly in WT and decreased significantly in BK-KO cells compared to low $[GLU]_{ex}$ (2.0 mM) (*Figure 4—figure supplement 1A*). Because $F_OF_1$-ATP-synthase (ATP Synthase) may hydrolyze ATP in an attempt to maintain $\Delta\Psi_{mito}$, these results strongly suggest that the forward (ATP producing) vs reverse (ATP consuming) mode of the ATP synthase is affected by the $BK_{Ca}$ status of the cells (*Naguib et al., 2018*).

To unravel the substrate dependency of these BCCs for maintaining their energy homeostasis in dependency of $BK_{Ca}$, we measured the mitochondrial ATP concentration ($[ATP]_{mito}$) using a genetically encoded ATP sensor targeted to the mitochondrial matrix, mtAT1.03 (*Imamura et al., 2009*). Mitochondrial ATP reportedly responds most dynamically to energy metabolism perturbations (*Depaoli et al., 2018*). To assess the sources of ATP in the cells, we either deprived the cells of $[GLU]_{ex}$, or administered Oligomycin-A to inhibit the ATP-synthase (*Depaoli et al., 2018*; *Figure 4E–J* and *Figure 4—figure supplement 1B—G*). In MMTV-PyMT cells, glucose removal as well as ATP-synthase inhibition reduced $[ATP]_{mito}$, albeit the effect was more pronounced upon $[GLU]_{ex}$ depletion (*Figure 4E and F* and *Figure 4—figure supplement 1B and C*). Remarkably, paxilline, but not iberiotoxin treatment reduced the glucose, and increased ATP-synthase dependency of MMTV-PyMT WT cells for maintaining $[ATP]_{mito}$ (*Figure 4E and F* and *Figure 4—figure supplement 1B and C*). These observations were confirmed in MDA-MB-453 cells, even though (i) these cells showed an overall higher

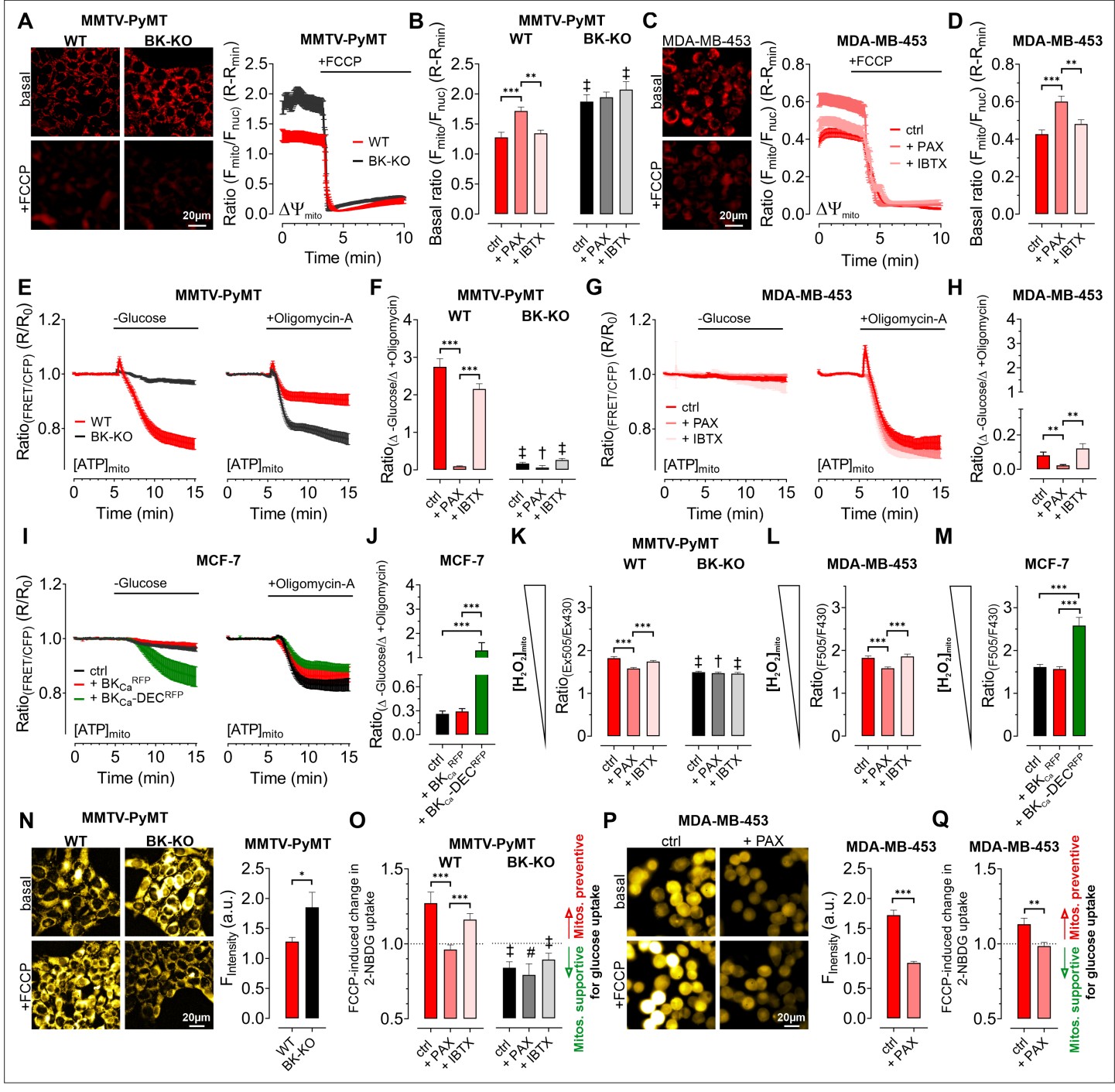

**Figure 4.** Expression of BK$_{Ca}$ modulates mitochondrial function and glucose uptake of BCCs. (**A – D**) Representative fluorescence images and -ratios ($F_{mito}/F_{nuc}$) over-time (**A, C**), and corresponding statistics ± SEM (**B, D**) representing $\Delta\Psi_{mito}$ of TMRM-loaded MMTV-PyMT WT and BK-KO (**A, B**) and MDA-MB-453 cells (**C, D**) under basal conditions (**A, C**, upper images) and upon administration of FCCP for mitochondrial depolarization (**A, C**, lower images). (**E – J**) [ATP]$_{mito}$ dynamics ± SEM over-time of MMTV-PyMT WT and BK-KO cells (**E**), MDA-MB-453 cells (**G**) and MCF-7 cells (**I**) in response to extracellular glucose removal (left panels) or upon administration of Oligomycin-A (right panels). (**F, H**) and (**J**) show changes of [ATP]$_{mito}$ induced by glucose removal to Oligomycin-A administration ± SEM, under control conditions, or in the presence of paxilline or iberiotoxin (**F, H**), or upon expression of BK$_{Ca}^{RFP}$ or BK$_{Ca}$-DEC$^{RFP}$ (**J**). (**K – M**) Basal mitochondrial H$_2$O$_2$ concentrations ± SEM of MMTV-PyMT WT (**K**, left), BK-KO (**K**, right), MDA-MB-453 (**L**) and MCF-7 cells (**M**), either under control conditions, in the presence of paxilline or iberiotoxin (**K, L**), or upon expression of BK$_{Ca}^{RFP}$ or BK$_{Ca}$-DEC$^{RFP}$ (**M**). (**N, P**) Representative fluorescence wide-field images (left) and corresponding statistics ± SEM (right) of MMTV-PyMT WT (**N**, left images and red bars) and BK-KO cells (**N**, right images and black bars) or MDA-MB-453 cells (**P**) incubated with 2-NBDG, either in the absence (upper images) or presence of FCCP (lower images). (**O, Q**) Average ± SEM of FCCP induced change in 2-NBDG uptake of MMTV-PyMT WT (**O**, left) and BK-

*Figure 4 continued on next page*

*Figure 4 continued*

KO cells (**O**, right), or MDA-MB-453 cells (**Q**) either under control conditions, or in the presence of paxilline or iberiotoxin. Values above 1 indicate that mitochondria prevent, values below 1 that mitochondria support glucose uptake. N (independent experiments) / n (cells analyzed) = (**A, B**): 4/75 WT ctrl, 4/90 WT +PAX, 4/86 WT +IBTX, 4/91 BK-KO ctrl, 4/89 BK-KO +PAX, 4/100 BK-KO +IBTX, (**C, D**): 4/113 ctrl, 4/97+PAX, 4/103+IBTX, (**E, F**): [-Glucose]: 8/55 WT ctrl, 6/45 WT +PAX, 7/27 WT +IBTX, 8/65 BK-KO ctrl, 6/57 BK-KO +PAX, 7/28 BK-KO +IBTX, [+Oligomycin-A]: 11/52 WT ctrl, 7/53 WT +PAX, 7/34 WT +IBTX, 8/87 BK-KO ctrl, 6/35 BK-KO +PAX, 5/45 BK-KO +IBTX. (**G, H**): [-Glucose]: 5/14 ctrl, 3/13+PAX, 5/13+IBTX, [+Oligomycin-A]: 5/33 ctrl, 3/21+PAX, 8/27+IBTX, (**I, J**): [-Glucose]: 6/48 ctrl, 5/23+BK$_\text{Ca}^\text{RFP}$, 5/20+BK$_\text{Ca}$-DEC$^\text{RFP}$, [+Oligomycin-A]: 5/27 ctrl, 5/23+BK$_\text{Ca}^\text{RFP}$, 5/37+BK$_\text{Ca}$-DEC$^\text{RFP}$, (**K**): 3/33 WT ctrl, 4/51 WT +PAX, 4/54 WT +IBTX, 4/55 BK-KO ctrl, 4/51 BK-KO +PAX, 4/54 BK-KO +IBTX, (**L**): 4/31 ctrl, 4/39+PAX, 4/31+IBTX, (**M**): 4/29 ctrl, 4/17+BK$_\text{Ca}^\text{RFP}$, 4/21+BK$_\text{Ca}$-DEC$^\text{RFP}$, (**N – Q**): 4 for all. *p≤0.05, **p≤0.01, ***p≤0.001, Kruskal-Wallis test followed by Dunn's MC test (**B, D, F, H, J, K, O**), Brown-Forsythe and Welch ANOVA test followed by Games-Howell's MC test (**L, M**), Mann-Whitney test (**N**), Unpaired t-test (**P**) or Welch's t-test (**Q**). #p≤0.05, †p≤0.01, ‡p≤0.001, to respective WT condition, Mann-Whitney test (**B, F**)  +PAX and+IBTX in (**K**) ctrl in (**O**), Unpaired t-test (ctrl in **K**)  +PAX and+IBTX in (**O**).

The online version of this article includes the following source data and figure supplement(s) for figure 4:

**Source data 1.** Numerical values underlying the data shown in *Figure 4*.

**Figure supplement 1.** BK$_\text{Ca}$ modulates cellular substrate dependency for maintaining [ATP]$_\text{mito}$ and reverses F$_\text{O}$F$_1$ ATP-synthase.

**Figure supplement 1—source data 1.** Numerical values underlying the data shown in *Figure 4*.

dependency on the ATP-synthase, and (ii) the paxilline-sensitivity of [ATP]$_\text{mito}$ maintenance was much less pronounced (*Figure 4G and H* and *Figure 4—figure supplement 1D and E*).

To validate these findings in a BK$_\text{Ca}$ minimal to depleted model, these experiments were performed with MCF-7 cells either expressing exclusively RFP as control, BK$_\text{Ca}^\text{RFP}$, or BK$_\text{Ca}$-DEC$^\text{RFP}$ (*Figure 4I and J*). Expression of BK$_\text{Ca}$-DEC$^\text{RFP}$, but not BK$_\text{Ca}^\text{RFP}$, resulted in a high dependency on [GLU]$_\text{ex}$ to maintain [ATP]$_\text{mito}$ while the [ATP]$_\text{mito}$ rundown in the presence of Oligomycin-A was identical for both BK$_\text{Ca}$ splice variants. This suggests that BK$_\text{Ca}$-DEC$^\text{RFP}$ specifically triggers a high [GLU]$_\text{ex}$ sensitivity and simultaneously an independency on ATP derived from ATP-synthase for maintaining [ATP]$_\text{mito}$ as demonstrated by the ratio of the respective changes under these experimental conditions (*Figure 4I and J* and *Figure 4—figure supplement 1F and G*). These findings suggest that intracellular (mitochondrial) BK$_\text{Ca}$ contributes to the metabolic reprogramming of BCCs.

In an extension to extracellular flux analyses, single time-point LC-MS metabolomics (*Figure 3*), and high-resolution live-cell imaging experiments (*Figure 4A–J*), pointing to a BK$_\text{Ca}$-dependent 'oncometabolic' phenotype, we assessed the concentrations of mitochondrial hydrogen peroxide ([H$_2$O$_2$]$_\text{mito}$) using mitoHyPer3 (*Bilan et al., 2013*), a genetically encoded fluorescent indicator for monitoring H$_2$O$_2$ in the mitochondrial matrix (*Figure 4K–M*). These experiments demonstrated increased levels of [H$_2$O$_2$]$_\text{mito}$ in MMTV-PyMT WT (*Figure 4K*) and MDA-MB-453 cells (*Figure 4L*), or specifically upon BK$_\text{Ca}$-DEC$^\text{RFP}$ expression in MCF-7 cells (*Figure 4M*). Excessive reactive oxygen species (ROS) synthesis is caused by uncoupling of the respiratory chain and it is considered as an indicator of mitochondrial stress, which, among other reasons, promotes mutagenesis and BCC progression. In this regard, mitoBK$_\text{Ca}$ may serve as an "uncoupling" protein (*Gałecka et al., 2021*), triggering ATP synthase to operate in reverse mode to consume instead of producing ATP, thereby counteracting the dissipation of the proton gradient and consequently the loss of ΔΨ$_\text{mito}$. Consequently, the lack of ATP must be compensated, e.g. by accelerating glycolysis. To investigate this assumption, we performed glucose uptake measurements using 2-NBDG, a fluorescent glucose analogue, which is taken up via glucose transporters (GLUTs), phosphorylated by hexokinase (HK) isoforms to generate 2-NBDG-6-phosphate and subsequently remains within the cell (*Bischof et al., 2021*; *Figure 4—figure supplement 1H and I*). Unexpectedly, we found that MMTV-PyMT BK-KO cells showed a higher rate of glucose uptake and phosphorylation compared to WT cells under basal conditions (*Figure 4N*). To unravel the role of mitochondria in contributing to glucose uptake by supplying ATP to HKs, we next performed these experiments in the presence of FCCP, which disrupts mitochondrial ATP production (*Losano et al., 2017*). If ATP-synthase works in forward mode, FCCP treatment should reduce or even prevent oxidative ATP production, and, subsequently, ATP-dependent 2-NBDG phosphorylation (*Figure 4—figure supplement 1H*). Contrary, if ATP-synthase works in reverse mode, it may compete with HKs for ATP, and FCCP treatment should abolish this competition, leading to increased 2-NBDG phosphorylation in BCCs (*Figure 4—figure supplement 1I*). Interestingly, our experiments unveiled increased 2-NBDG uptake in MMTV-PyMT WT and reduced uptake in BK-KO cells upon mitochondrial depolarization (*Figure 4N* and *Figure 4—figure supplement 1J and K*). To facilitate the interpretation,

we calculated the FCCP-induced change in 2-NBDG uptake. These analyses revealed that mitochondria are less effective in assisting 2-NBDG uptake and phosphorylation in MMTV-PyMT WT cells, while they 'support' these processes in BK-KO cells under control conditions (*Figure 4O*). To test whether the changes in 2-NBDG uptake are sensitive to pharmacological modulation of $BK_{Ca}$, we performed a similar set of experiments in the presence of paxilline or iberiotoxin. While iberiotoxin treatment did not have any effect, paxilline treatment shifted the activity of the ATP-synthase to the ATP-supplying 'forward' mode (*Figure 4O* and *Figure 4—figure supplement 1J, K*). Comparable effects were obtained in MDA-MB-453, despite paxilline treatment reduced basal accumulation of 2-NBDG in these cells (*Figure 4P and Q* and *Figure 4—figure supplement 1L*). Overall, the experiments performed so far suggest that mitochondria rather consume than generate ATP if $BK_{Ca}$ was functionally expressed intracellularly in BCCs.

## $BK_{Ca}$ locates functionally in the IMM of murine and human BCCs

So far, our data suggest an important contribution of intracellularly located $BK_{Ca}$, possibly mito$BK_{Ca}$, in reprogramming cancer cell metabolism. Thus, we applied an electrophysiological approach to provide functional evidence for endogenous mito$BK_{Ca}$. Single-channel patch-clamp experiments were conducted using mitoplasts isolated from MMTV-PyMT WT and BK-KO, MDA-MB-453, and MCF-7 cells. In MDA-MB-453- and MMTV-PyMT WT-derived mitoplasts we indeed detected channels of large conductance (*Figure 5A* and *Figure 5—figure supplement 1A*). For these channels, the open probability did not significantly vary with the voltage (*Figure 5B*). Only at very negative and positive voltages of approximately –150 mV and +150 mV differences in the open probabilities were observed (*Figure 5C*). The bursts of single-channel openings showed an average conductance of 212±2 pS (*Figure 5D*). This large conductance and the sensitivity towards $Ca^{2+}$- (*Figure 5E and F* and *Figure 5—figure supplement 1B*) and paxilline (*Figure 5E and F* and *Figure 5—figure supplement 1C*) pointed to mito$BK_{Ca}$ that exhibits the pharmacologic characteristics of canonical $BK_{Ca}$ channels present at the PM. Overall, our electro-pharmacological experiments unveiled 3 different classes of channels with either small (≤100 pS), medium (≤150 pS), or large (~210 pS) conductance, where the latter conductance corresponds to mito$BK_{Ca}$ (*Liu et al., 1999*). The lower conductance values may represent mitochondrial $IK_{Ca}$, $SK_{Ca}$, or $K_{ATP}$ channel activity, but this was not investigated further in our study. Notably, mito$BK_{Ca}$ was detected at a frequency of 15% in 210 patches of MDA-MB-453 mitoplasts and at a lower frequency of 6% in 127 mitoplast patches of MMTV-PyMT WT cells. These numbers may, however, underestimate the actual abundance of the channel, as only a small area of the mitoplast is examined with each patch. Importantly, this channel was absent in 76 mitoplast patches from MMTV-PyMT BK-KO and 58 mitoplast patches from MCF-7 cells (*Figure 5G*). These findings provide evidence that the molecular entity for the $K^+$ channel derives from the nuclear *Kcnma1* gene, which is ablated in the MMTV-PyMT BK-KO.

These experiments were further corroborated by immunoblotting experiments using whole-cell lysates and subcellular homogenates of different purity. We detected $BK_{Ca}$, the $Na^+/K^+$ ATPase as a marker of the PM, cytochrome c oxidase subunit IV (COXIV) as a marker of mitochondria and TMX1, which localizes to ER membranes in these samples (*Figure 5—figure supplement 1D*). A protein band corresponding to $BK_{Ca}$ was identified not only in whole-cell lysates, but also in isolated mitochondria. Importantly, the $Na^+/K^+$ ATPase was absent in the latter protein fraction, confirming the purity of the mitochondrial preparation (*Figure 5—figure supplement 1D*). These data confirm the presence of mito$BK_{Ca}$, potentially $BK_{Ca}$-DEC, in the utilized $BK_{Ca}$ proficient BCCs.

## mito$BK_{Ca}$ promotes the Warburg effect, triggers cellular $O_2$ independency and stimulates BCC proliferation

Based on the observed influence of mito$BK_{Ca}$ on glycolysis and mitochondrial metabolism, we addressed, whether the channel contributes to the Warburg effect, commonly observed in cancer cells (*Bischof et al., 2021*; *Warburg, 1924*). Therefore, we assessed the Warburg index (WI) by investigating the cytosolic lactate concentration ($[LAC]_{cyto}$) over-time using Laconic, a FRET-based lactate sensor (*San Martín et al., 2013*). $[LAC]_{cyto}$ was followed in response to either inhibiting mitochondrial metabolism by administration of $NaN_3$, a complex IV inhibitor, to stop pyruvate consumption (*Leary et al., 1998*), or upon subsequent inhibition of lactate secretion towards the ECM via monocarboxylate transporter 1 (MCT-1) using BAY-8002 (*Quanz et al., 2018*; *Figure 6A and B*). Indeed, MMTV-PyMT

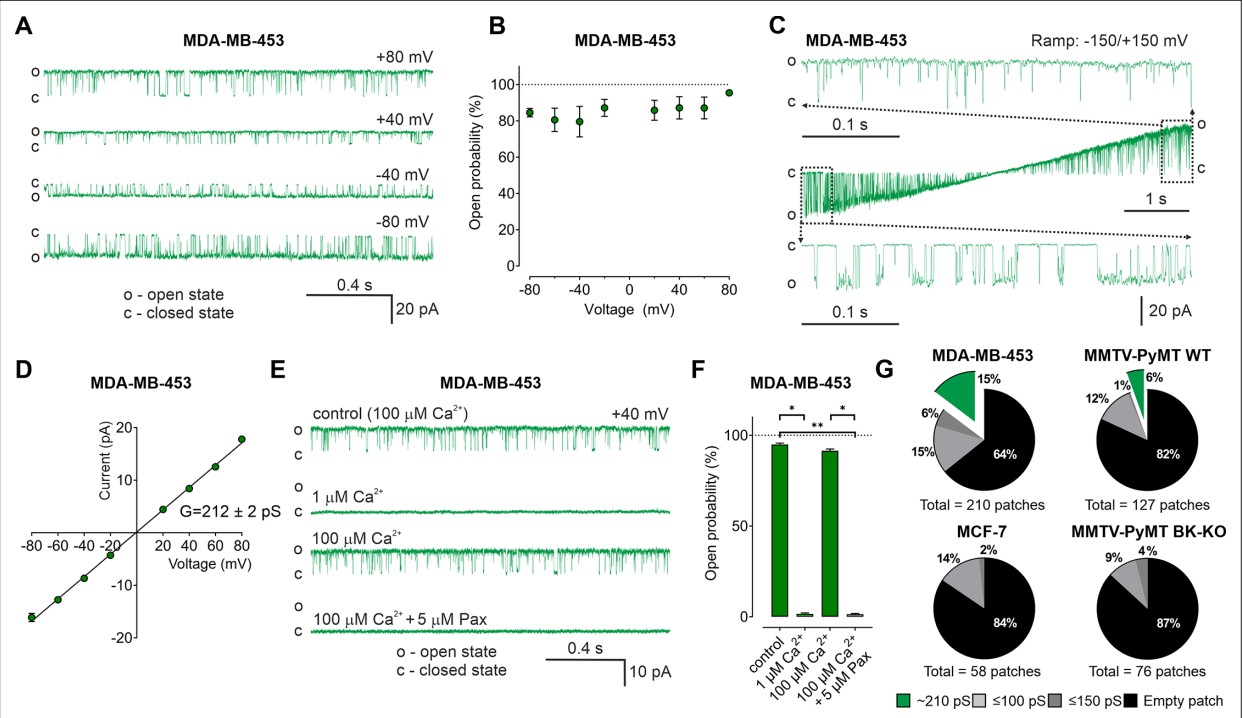

**Figure 5.** BK$_{Ca}$ activity is present in the inner mitochondrial membrane (IMM) of BCCs. (**A**) Representative BK$_{Ca}$ single-channel recordings of the IMM of mitoplasts isolated from MDA-MB-453 cells using a symmetric 150/150 mM isotonic KCl solution containing 100 µM Ca$^{2+}$ at voltages ranging from –80 to +80 mV as indicated in the panel. (**B**) Open probability analysis of mitoBK$_{Ca}$ at different voltages received from experiments as performed in (**A**). N = 8. (**C**) Single-channel currents of the IMM of mitoplasts isolated from MDA-MB-453 cells recorded using a voltage ramp protocol ranging from −150 to +150 mV. Above and below the ramp are enlarged excerpts of the records shown in rectangles. (**D**) Current-voltage (**I–V**) plot based on single-channel recordings of MDA-MB-453 cells as performed in (**A**), using a symmetric 150/150 mM KCl isotonic solution containing 100 µM Ca$^{2+}$. N = 11. (**E, F**) Representative single-channel recordings of the IMM of mitoplasts isolated from MDA-MB-453 cells (**E**) and corresponding open probabilities at +40 mV in a symmetric 150/150 mM KCl isotonic solution under control conditions (100 µM Ca$^{2+}$), after reducing Ca$^{2+}$ to 1 µM, re-addition of 100 µM Ca$^{2+}$ and finally after application of 5 µM paxilline in the presence of 100 µM Ca$^{2}$. Data in (**F**) show average ± SEM. *p≤0.05, **p≤0.01 using Friedmann test followed by Dunn's multiple comparison test, n = 7. (**G**) Pie chart displaying the percentage of mitoBK$_{Ca}$ channel currents (green) possessing a conductance of ~210 pS, *versus* the total number of patch-clamp experiments performed using mitoplasts isolated from MDA-MB-453 cells (upper left), MMTV-PyMT WT cells (upper right), MCF-7 cells (lower left), and MMTV-PyMT BK-KO cells (lower right). Black segments represent empty patches, bright- and dark grey fraction demonstrate percentage of channels possessing smaller conductances of ≤100 pS and ≤150 pS, respectively. All recordings were low-pass filtered at 1 kHz. 'c' and 'o' indicate the closed- and open state of the channel, respectively.

The online version of this article includes the following source data and figure supplement(s) for figure 5:

**Source data 1.** Numerical values underlying the data shown in *Figure 5*.

**Figure supplement 1.** BK$_{Ca}$ is present in the inner mitochondrial membrane of MMTV-PyMT WT and MDA-MB-453 cells.

**Figure supplement 1—source data 1.** Original image of the Western Blot against TMX1.

**Figure supplement 1—source data 2.** Original image of the Western Blot against TMX1 with bands used in the Figure indicated.

**Figure supplement 1—source data 3.** Original image of the Western Blot against Na$^+$/K$^+$ ATPase.

**Figure supplement 1—source data 4.** Original image of the Western Blot against Na$^+$/K$^+$ ATPase with bands used in the Figure indicated.

**Figure supplement 1—source data 5.** Original image of the Western Blot against BK$_{Ca}$.

**Figure supplement 1—source data 6.** Original image of the Western Blot against BK$_{Ca}$ with bands used in the Figure indicated.

**Figure supplement 1—source data 7.** Original image of the Western Blot against COX IV.

**Figure supplement 1—source data 8.** Original image of the Western Blot against COX IV with bands used in the Figure indicated.

WT cells exhibited an increased WI compared to BK-KO cells under control conditions (*Figure 6C*), indicating that the presence of BK$_{Ca}$ favors lactate secretion rather than TCA-dependent utilization of pyruvate. Paxilline, but not iberiotoxin treatment reduced the WI in WT cells to the BK-KO level (*Figure 6C*). The WI profiles of MDA-MB-453 (*Figure 6D*) and MCF-7 cells (*Figure 6E*) showed the same sensitivity towards paxilline, while in the latter the expression of BK$_{Ca}$-DEC$^{RFP}$, but not BK$_{Ca}^{RFP}$,

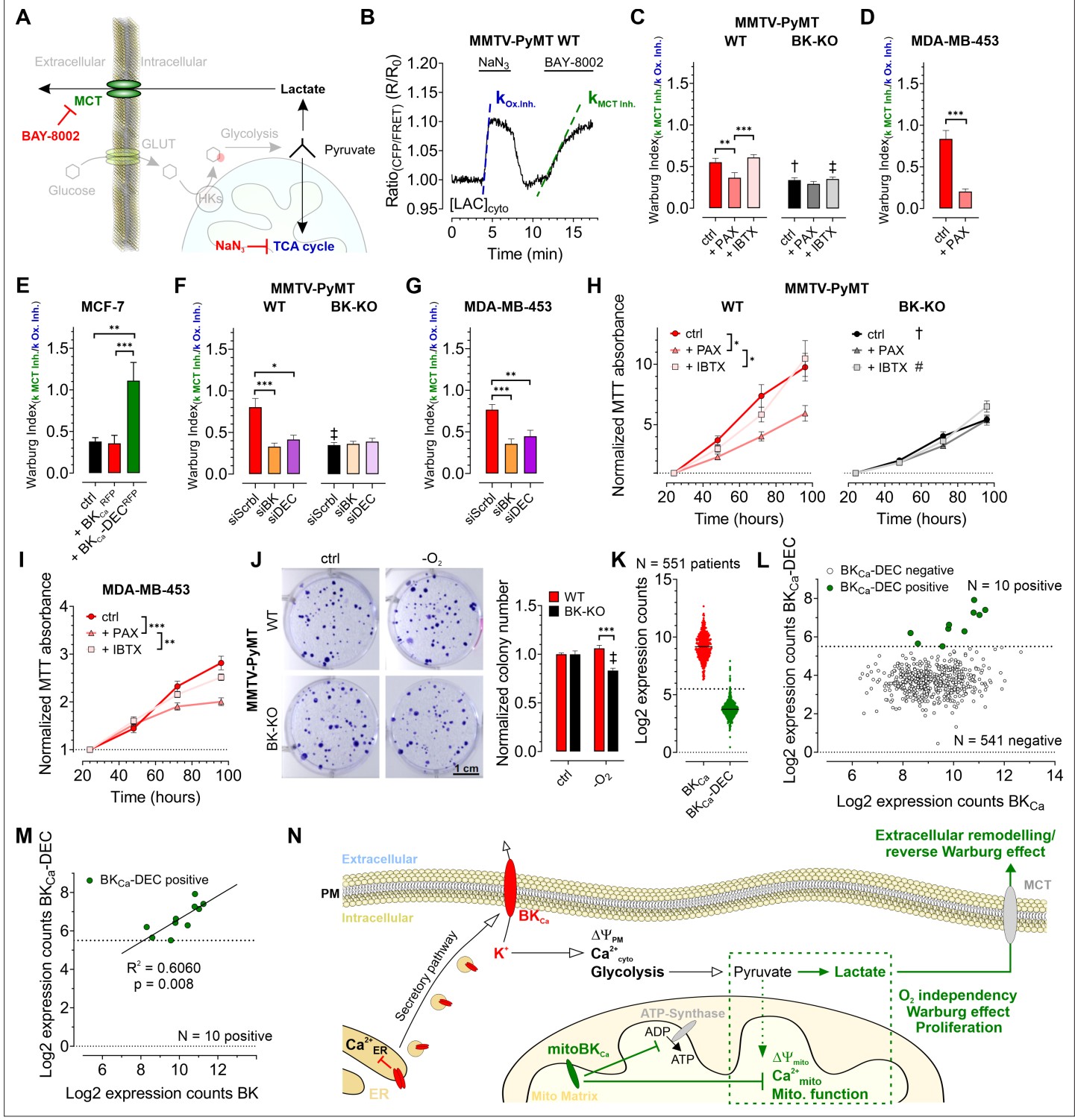

**Figure 6.** BK$_{Ca}$-DEC expression contributes to the metabolic remodeling and growth of murine and human BCCs and is present in primary tumor samples. (**A**) Schematic representation of the fate of glucose in glycolysis. The tricarboxylic acid (TCA) cycle or lactate secretion via monocarboxylate transporters (MCT) can be inhibited, either using NaN$_3$ or BAY-8002. GLUT: Glucose transporter, HKs: Hexokinases. (**B**) Representative cytosolic lactate concentration ([LAC]$_{cyto}$) of a MMTV-PyMT WT cell over-time in response to administration or removal of NaN$_3$ and BAY-8002 at time points indicated. Dashed lines indicate slopes taken for assessment of the 'Warburg index'. (**C – G**) Average Warburg indices ± SEM of MMTV-PyMT WT (**C, F**, left), MMTV-PyMT BK-KO (**C, F**, right), MDA-MB-453 cells (**D, G**) and MCF-7 cells (**E**) calculated from the experiments as shown in (**B**), either under control conditions, in the presence of paxilline or iberiotoxin (**C, D**), upon expression of BK$_{Ca}$$^{RFP}$ or BK$_{Ca}$-DEC$^{RFP}$ (**E**), or upon cell treatment with a scrambled

*Figure 6 continued on next page*

*Figure 6 continued*

siRNA (siScrbl), or siRNA against a common BK$_{Ca}$ sequence targeting all known splice variants (siBK), or a siRNA specifically designed to knockdown BK$_{Ca}$-DEC (siDEC) (**F, G**). (**H, I**) Normalized MTT absorbance over-time of MMTV-PyMT WT (**H**, left) and BK-KO cells (**H**, right), and MDA-MB-453 cells (**I**), either under control conditions, or in the presence of paxilline or iberiotoxin. (**J**), Representative images and corresponding statistics of colony formation assays using MMTV-PyMT WT or BK-KO cells in the presence or absence of O$_2$. (**K – N**) mRNA expression of BK$_{Ca}$ and BK$_{Ca}$-DEC as performed by Nanostring analysis of 551 BC patient samples. (**K**) Log2 expression counts of BK$_{Ca}$ and BK$_{Ca}$-DEC. The threshold for positive expression level was set to log2 = 5.5 (dashed line). (**L**) Log2 expression counts of BK$_{Ca}$-DEC blotted over the log2 expression counts of BK$_{Ca}$. 10 of the 551 patient samples showed expression of BK$_{Ca}$-DEC above the threshold of log2 = 5.5 (dashed line), whereas 541 patient samples were BK$_{Ca}$-DEC negative. (**M**) Correlation of the log2 expression counts of BK$_{Ca}$-DEC positive samples with the log2 expression counts of BK$_{Ca}$ in the primary human BC material. (**N**) Summarizing scheme of BK$_{Ca}$ in cancer cell homeostasis. N (independent experiments) / n (cells analyzed) = (**C**): 4/26 WT ctrl, 6/28 WT +PAX, 4/39 WT +IBTX, 4/17 BK-KO ctrl, 5/18 BK-KO +PAX, 4/27 BK-KO +IBTX, (**D**): 7/29 ctrl, 5/13+PAX, (**E**): 5/27 ctrl, 5/20+BK$_{Ca}$$^{RFP}$, 7/26 BK$_{Ca}$-DEC$^{RFP}$, (**F**): 5/22 WT siScrbl, 5/28 WT siBK, 4/26 WT siDEC, 5/24 BK-KO siScrbl, 5/24 BK-KO siBK, 5/29 BK-KO siDEC, (**G**): 5/21 siScrbl, 5/22 siBK, 5/19 siDEC, (**H – J**): 4 for all. *p≤0.05, **p≤0.01, ***p≤0.001, Kruskal-Wallis test followed by Dunn's MC test (**C, E, F, G, I**), One-Way ANOVA test followed by Tukey's MC test (**H**) or Mann-Whitney test (**D**). †p≤0.01, ‡p≤0.001 compared to respective WT condition, Unpaired t-test ctrl, (**C, J**) or Mann-Whitney test (+IBTX, **C, F**).

The online version of this article includes the following source data and figure supplement(s) for figure 6:

**Source data 1.** Numerical values underlying the data shown in *Figure 6*.

**Figure supplement 1.** Effects of siRNA treatment on expression of BK$_{Ca}$ and BK$_{Ca}$-DEC, and physiologic consequences of BK$_{Ca}$ inhibition in BCCs.

**Figure supplement 1—source data 1.** Numerical values underlying the data shown in *Figure 6*.

---

stimulated the WI. To validate the contribution of the different BK$_{Ca}$ isoforms, endogenous BK$_{Ca}$ transcripts in MMTV-PyMT and MDA-MB-453 cells were targeted using specific siRNAs targeting either the major BK$_{Ca}$ isoforms or specifically the DEC exon of BK$_{Ca}$ (*Figure 6—figure supplement 1A*). Cell treatment with the respective siRNAs reduced the expression of BK$_{Ca}$ or BK$_{Ca}$-DEC (*Figure 6—figure supplement 1B and C*). Interestingly, the knockdown of BK$_{Ca}$ interfered with the WI in these cells (*Figure 6F and G*). This effect was reproduced by specific silencing of the BK$_{Ca}$-DEC isoform (*Figure 6F and G*), indicating that BK$_{Ca}$-DEC-derived mitoBK$_{Ca}$ channels stimulate the Warburg effect in BCCs.

As glycolytic metabolites and ATP fuel cell proliferation, we investigated the proliferation rates of MMTV-PyMT WT, MMTV-PyMT BK-KO, and MDA-MB-453 cells over-time in the absence or presence of paxilline or iberiotoxin. Using MTT-assay, corroborating previous findings (*Mohr et al., 2022*), these analyses revealed faster proliferation of WT compared to BK-KO cells under control conditions (*Figure 6H*). Interestingly, paxilline, but not iberiotoxin treatment, reduced the proliferation of MMTV-PyMT WT to the level of BK$_{Ca}$-deficient cells (*Figure 6H*). These effects were also observed in MDA-MB-453 cells (*Figure 6I*). Since the MTT assay is also a readout for mitochondrial function, we additionally performed immunofluorescence-based detection of Ki-67, a frequently used marker of cell proliferation (*Dowsett et al., 2011*). This approach confirmed the MTT data (*Figure 6—figure supplement 1D and F*), establishing intracellular BK$_{Ca}$, possibly mitoBK$_{Ca}$, as key-player in mediating the metabolic rewiring. Next, we studied whether the increased WI affects the proliferation rate of these BCCs at low O$_2$ tension (*Nazemi and Rainero, 2020*). Colony formation assays (CFAs) revealed an increased hypoxic resistance of MMTV-PyMT WT compared to BK-KO cells, as the number and size of colonies lacking BK$_{Ca}$ was reduced upon O$_2$-deprivation (*Figure 6J* and *Figure 6—figure supplement 1G*). In sum, these data demonstrate a malignancy-promoting effect of mitoBK$_{Ca}$ in BCC.

## mitoBK$_{Ca}$ is of potential clinical relevance

To determine the clinical relevance of our findings, we investigated whether BK$_{Ca}$-DEC transcripts are present in primary BC tissue. Therefore, the mRNA expression of BK$_{Ca}$ and BK$_{Ca}$-DEC was analyzed by nanostring analysis in bulk tumor biopsies isolated from 551 BC patients. Remarkably, all 551 samples tested positive for BK$_{Ca}$ mRNA expression, with 10 of the samples showing significant expression of BK$_{Ca}$-DEC above the log2 expression threshold of 5.5 (*Figure 6K and L*). Importantly, if BK$_{Ca}$-DEC was expressed, the expression of BK$_{Ca}$ well correlated with the expression of BK$_{Ca}$-DEC ($R^2$=0.6060, p=0.008) (*Figure 6M*).

In sum, our experiments emphasize the presence of BK$_{Ca}$ in different intracellular organelles including the ER and vesicles of the secretory pathway, yielding its PM localization. At these sites, BK$_{Ca}$ modulates the Ca$^{2+}$ homeostasis and regulates $\Delta\Psi_{PM}$. K$^+$ efflux across the PM may additionally affect glycolysis. Importantly, functionally relevant BK$_{Ca}$ also locates in the IMM of BCCs, promoting,

presumably by the $K^+$ accumulation in the matrix following channel activation, $\Delta\Psi_{mito}$ depolarization and consequently ATP-synthase activity in reverse mode as well as a depletion of $[Ca^{2+}]_{mito}$. These profound ionic and bioenergetic changes ultimately trigger the proliferation of BCCs in a low oxygen environment, as found in solid tumors (*Figure 6N*). Taken together, functional expression of mitoBK$_{Ca}$ could possibly denote a prognostic or therapeutic marker for BC patients, and its pharmacologic modulation could represent a novel anti-cancer treatment strategy.

## Discussion

Here, we demonstrate for the first time that BK$_{Ca}$-DEC (mitoBK$_{Ca}$) is functionally expressed in BCCs. BK$_{Ca}$-DEC modulates BCC metabolism, stimulates the Warburg effect, and accelerates cell proliferation rates in the presence and absence of $O_2$. These tumor- and malignancy-promoting effects were sensitive to BK$_{Ca}$ inhibition using the cell-permeable BK$_{Ca}$ inhibitor paxilline (*Zhou and Lingle, 2014*), but not the cell-impermeable blocker iberiotoxin (*Candia et al., 1992*), indicating that intracellular BK$_{Ca}$, presumably mitoBK$_{Ca}$, mediates malignant BCC behaviors and tumor development.

In line with recent single-cell RNA sequencing data of 26 primary breast tumors (*Wu et al., 2021a*), we found high transcript levels for BK$_{Ca}$ throughout the analyzed BC samples, in addition to BK$_{Ca}$-DEC, albeit in a much smaller subset of BC. The analyzed BC samples were, however, all positive for hormone receptors. Whether BK$_{Ca}$ expression is different in hormone receptor negative specimens hence needs to be further investigated. Further, we confirmed functional BK$_{Ca}$ expression in the PM of MMTV-PyMT WT and MDA-MB-453 cells, while MMTV-PyMT BK-KO and MCF-7 cells showed no or very low PM BK$_{Ca}$ currents. Therefore, MCF-7 cells represented a suitable model to investigate the effects of BK$_{Ca}$ over-/expression on the metabolic homeostasis of human BCC. If the low BK$_{Ca}$ expression correlates with hormonal receptor status or alternatively with human epidermal growth factor 2 (HER2) expression levels must be clarified by future studies. Interestingly, however, recent data have shown that BK$_{Ca}$ is overexpressed in triple-negative BC, a fact that led the authors to draw similar conclusions to ours, that BK$_{Ca}$ may represent a novel anti-cancer treatment strategy for selected BC patients (*Sizemore et al., 2020*). Nevertheless, similarly to cardiac myocytes (CM), expression of BK$_{Ca}$-DEC yielded mitochondrial localization of BK$_{Ca}$-DEC, although its abundance in the IMM appeared less pronounced compared to CM (*Singh et al., 2013*). While a direct comparison is difficult as plasmid transfections may have caused unexpected effects, our finding that BK$_{Ca}$-DEC$^{RFP}$ caused significantly reduced currents across the PM compared to BK$_{Ca}$$^{RFP}$ putatively confirm its increased intracellular abundance.

Based on the potential impact of BK$_{Ca}$ on the cellular $\Delta\Psi_{PM}$ and ion balance (*Burgstaller et al., 2022a*), we conducted an in-depth investigation of the (sub)cellular $Ca^{2+}$ homeostasis. Across the BCCs examined, we observed that functional BK$_{Ca}$ expression modulated $[Ca^{2+}]_{cyto}$ dynamics. These alterations showed, however, differential sensitivities to BK$_{Ca}$ inhibitors in MMTV-PyMT WT and MDA-MB-453 cells. While basal $[Ca^{2+}]_{cyto}$ in MDA-MB-453 cells was reduced by both, paxilline and iberiotoxin treatment, MMTV-PyMT WT cells were only sensitive to paxilline. Examination of $[Ca^{2+}]_{ER}$ confirmed the results from $[Ca^{2+}]_{cyto}$, as $[Ca^{2+}]_{ER}$ levels increased with paxilline and iberiotoxin in MDA-MB-453, but only upon paxilline exposure in MMTV-PyMT WT cells, suggesting differential effects of intracellular and PM-localized BK$_{Ca}$ on $Ca^{2+}$ handling in these cells. Interestingly, in MCF-7 cells, both BK$_{Ca}$ splice variants mediated the opposing effects on the basal $[Ca^{2+}]_{cyto}$ and $[Ca^{2+}]_{ER}$ levels, which either in- or decreased, respectively, upon transient BK$_{Ca}$$^{RFP}$ or BK$_{Ca}$-DEC$^{RFP}$ expression. This is expected as the transitory hyperpolarization and the efflux of $K^+$ due to the opening of PM BK$_{Ca}$ provides the driving force for $Ca^{2+}$ entry into cytoplasm, while a BK$_{Ca}$-mediated $K^+$ increase within the ER, presumably through PM-directed channels crossing the ER membrane in the secretory pathway, would oppose the $Ca^{2+}$ refilling (*Burgstaller et al., 2022a*). Moreover, these results are in line with the higher proliferative capability of BK$_{Ca}$ proficient BCCs due to the manifold roles of $Ca^{2+}$ as second messenger (*Burgstaller et al., 2022a*). Importantly, however, $[Ca^{2+}]_{mito}$ of BK$_{Ca}$ proficient MMTV-PyMT WT and MDA-MB-453 cells was exclusively sensitive for paxilline, and it was specifically affected by the expression of BK$_{Ca}$-DEC$^{RFP}$, but not by BK$_{Ca}$$^{RFP}$, in MCF-7 cells, suggesting that this $Ca^{2+}$ pool is exclusively controlled by intracellular BK$_{Ca}$.

Subcellular $Ca^{2+}$ alterations could reportedly alter cellular bioenergetics, as $Ca^{2+}$ directly regulates metabolic enzymes and activities (*Rossi et al., 2019*). Indeed, extracellular flux analysis, LC-MS-based metabolomics and fluorescence-based live-cell imaging confirmed, that the observed alteration in subcellular $Ca^{2+}$ homeostasis caused by BK$_{Ca}$, especially mitoBK$_{Ca}$, has severe effects on cell metabolism.

Our data further emphasize, that the presence of mitoBK$_{Ca}$, as confirmed by mitoplast patch-clamp and western blot analyses of isolated mitochondria, depolarizes BCC mitochondria, which is in line with a previous study showing an impact of mitoBK$_{Ca}$ activation on $\Delta \Psi_{mito}$ (*Kicinska et al., 2016*). mitoBK$_{Ca}$-dependent depolarization of $\Delta \Psi_{mito}$ in turn triggers cellular glucose dependency and reverses the activity of the mitochondrial ATP-synthase to consume ATP for restoring $\Delta \Psi_{mito}$. Finally, BK$_{Ca}$-DEC-derived mitoBK$_{Ca}$ channels promote the Warburg effect and ultimately stimulated proliferation rates of BCCs. Our data fit earlier findings from glioma cells, where an $O_2$-sensitivity of mitoBK$_{Ca}$ was observed, which probably increased the hypoxic resistance of these cancer cells (*Gu et al., 2014*). It may seem contradictory that mitoBK$_{Ca}$ is highly expressed in CM, which are among one of the most oxidative cell types known. It is, however, important to mention, that CM, under physiologic conditions, do not show a metabolic Warburg setting, which is a common phenomenon of cancer cells propelling their $O_2$ independency, due to the hypoxic microenvironment. Moreover, it was demonstrated recently that the absence of (mito)BK$_{Ca}$ does not alter physiologic cardiac function. Only upon induction of ischemia and reperfusion injury, a lack of (mito)BK$_{Ca}$ promoted the susceptibility of the heart to cell death, resulting in increased infarction size (*Frankenreiter et al., 2017*). Hence, it can be concluded that (mito)BK$_{Ca}$ only played a role under conditions where mitochondria are, due to the absence of $O_2$, not properly functioning. Taking this view into account, the results derived from CM are consistent with our findings in BCCs, as (mito)BK$_{Ca}$ mediates the resistance to hypoxic stress to BCC.

Excessive production and release of lactate, a hallmark of the Warburg effect, leads to extracellular acidification, subsequently creating a microenvironment that promotes tumorigenesis and metastasis as well as the resistance to anti-tumor immune responses and therapy (*de la Cruz-López et al., 2019*; *Nazemi and Rainero, 2020*; *Wu et al., 2021b*). Extracellular K$^+$ [K$^+$]$_{ex}$, in turn, accumulating within the necrotic core of solid tumors, was shown to interfere with effector T-cell function triggering immune escape of cancer cells (*Eil et al., 2016*). To elucidate whether mitoBK$_{Ca}$ directly contributes to lactate-induced tumor aggressiveness or [K$^+$]$_{ex}$, live-cell imaging of extracellular metabolites in 3D BCC models should be applied in future investigations (*Burgstaller et al., 2022b*; *Burgstaller et al., 2021b*).

Finally, our results demonstrate for the first time that BK$_{Ca}$-DEC transcripts are present in human BC biopsies. Although only a small proportion of patients was positive for BK$_{Ca}$-DEC expression, this finding could be of considerable clinical relevance considering the link between mitoBK$_{Ca}$ function and BCC metabolism. Importantly, the design of our study likely underestimates the incidental number of BK$_{Ca}$-DEC positive BC, as (i) only hormone-receptor positive BC specimens were included, and (ii) bulk-tumor mRNA was analyzed, hampering the detection of low-abundant or tightly regulated transcripts against a strong background of non-cancer cells present in bulk tumor tissues. Finally, due to the small number of positive hits (N = 10 positive *versus* N = 541 BK$_{Ca}$-DEC negative specimens) and the lack of (sufficient) follow-up information in some of these cases, we are currently unable to correlate BK$_{Ca}$-DEC expression with, for example, treatment response or survival. Thus, future studies are warranted to show how the tumor's BK$_{Ca}$-DEC status can help to predict or therapeutically improve standard chemo-endocrine treatments. The rather low abundance of BK$_{Ca}$-DEC in the clinical samples, however, is in agreement with our mitoplast patch clamp experiments with mitoBK$_{Ca}$-mediated K$^+$ currents being detected at frequencies between 6 to 15%, suggesting that either a small proportion of mitochondria express functional mitoBK$_{Ca}$, or that the abundance of the channel per mitochondrion is low, requiring sensitive mechanistic approaches for its detection.

In summary, our data highlight a potentially druggable mitoBK$_{Ca}$ isoform in BCCs, whose molecular entity is mainly formed by the *Kcnma1* encoded BK$_{Ca}$-DEC splice variant. This channel promotes metabolic alterations in cancer cells, even under low-oxygen conditions, which may ultimately be of clinical interest for new anti-cancer therapies.

## Materials and methods
### Buffers and solutions

If not otherwise stated, all chemicals were purchased from Carl Roth GmbH, Karlsruhe, Germany.

Buffers used in this study comprised the following:

Physiologic buffer for single-cell live recordings contained (in mM): 138 NaCl, 5 KCl, 2 CaCl$_2$, 1 MgCl$_2$, 2 glucose, 10 HEPES, pH set to 7.4 with NaOH. No glucose was added during glucose removal

experiments, while glucose was increased to 25 mM to investigate glucose dependency of the mitochondrial membrane potential. 10 mM $CaCl_2$ instead of 2 mM $CaCl_2$ were added to obtain 10Ca buffer. 0.1 mM Ethylene glycol bis(2-aminoethylether)-N, N, N', N'-tetra acetic acid (EGTA; Sigma Aldrich Chemie GmbH, Taufkirchen, Germany) instead of 2.0 mM $CaCl_2$ was added to obtain $Ca^{2+}$-free buffer. For 0 mM $K^+$ buffer, 5 mM KCl was replaced by 5 mM NaCl. For 300 mM $K^+$ buffer, 300 mM KCl was added instead of 5 mM $K^+$ and addition of NaCl was omitted. The following compounds were added to yield the following final concentration: 3 µM Oligomycin-A, 200 nM Carbonyl cyanide-p-trifluoromethoxyphenylhydrazone (FCCP), 5 µM paxilline all from (Santa Cruz Biotechnology, Dallas, USA), 30 nM iberiotoxin (Selleckchem, Planegg, Germany), 5 mM $NaN_3$, 3 µM BAY-8002, 15 µM 2,5-Di-(t-butyl)–1,4-hydroquinone (BHQ) (all from Sigma Aldrich Chemie GmbH), 100 µM adenosine-5'-triphosphate (ATP), 5 µM ionomycin (Alomone Labs, Jerusalem, Israel), 15 µM gramicidin (Sigma-Aldrich Chemie GmbH). For $H_2O$ insoluble compounds, final DMSO concentration in the buffer did not exceed 0.1%.

Intracellular buffer used for whole-cell patch-clamp experiments contained (in mM): 130 K-Gluconate, 5 KCl, 2 Mg-ATP, 0.1 $CaCl_2$, 0.2 $Na_2$-GTP, 0.6 EGTA, 5 HEPES, pH = 7.2 with KOH.

Cell equilibration buffer fluorescence microscopy-based contained (in mM): 135 NaCl, 5 KCl, 2 $CaCl_2$, 1 $MgCl_2$, 2.6 $NaHCO_3$, 0.44 $KH_2PO_4$, 0.34 $Na_2HPO_4$, 10 glucose, 10 HEPES, 2 GlutaMAX, 1 sodium pyruvate, with 1 x MEM amino acids and 1 x MEM vitamins added (both Thermo Fisher Scientific). pH was adjusted to 7.4 using NaOH.

Buffers used for mitochondrial isolation, mitoplast preparation, and single channel patch-clamp comprised the following: The preparation solution contained (in mM): 250 sucrose, 5 HEPES, pH = 7.2. The mitochondrial storage buffer contained (in mM): 150 KCl, 0.1 $CaCl_2$, 20 HEPES, pH = 7.2. The hypotonic buffer contained (in mM) 0.1 $CaCl_2$, 5 HEPES, pH = 7.2. The hypertonic buffer contained (in mM): 750 KCl, 0.1 $CaCl_2$, 30 HEPES, pH 7.2. Low-$Ca^{2+}$ solution (1 µM $Ca^{2+}$) contained (in mM): 150 KCl, 1 EGTA, 0.752 $CaCl_2$, 10 HEPES, pH = 7.2.

## Cell culture and transfection

Mouse mammary tumor virus polyoma middle T antigen (MMTV-PyMT) cells were isolated from tumors of MMTV-PyMT transgenic FVB/N WT (*Kcnma1* proficient) or BK-KO (*Kcnma1* deficient) mice (Kcnma1^tm1Ruth, MGI ID 3050114). Tumor growth in vivo and biopsies were authorized by the local ethics *Committee for Animal Research* (Regierungspräsidium Tübingen (PZ1/16, PZ2/17, PZ4/20 G), Germany), and were performed in accordance with the *German Animal Welfare Act*. Animals were kept on a 12 hr light/ dark cycle under temperature- and humidity-controlled conditions with unlimited access to food (Altromin, Lage, Germany) and water. MMTV-PyMT cells used in this study were isolated from three to seven different female breast-cancer bearing WT and 3–4 different female breast-cancer bearing BK-KO animals at an age of ~12–14 weeks. Upon dissection, tumors were carefully minced into pieces using atraumatic forceps, lysed by 1 mg mL$^{-1}$ Collagenase-D (Roche, Basel, Switzerland) for 10 min, and cultured as follows: Cells were grown in modified improved minimal essential medium (IMEM) supplemented with 5% fetal bovine serum (FBS), 1 mM sodium pyruvate and 100 U mL$^{-1}$ penicillin and 100 µg mL$^{-1}$ streptomycin (all purchased from Thermo Fisher Scientific) at 37 °C and 5% $CO_2$ in a humidified incubator. Fibroblasts were removed by exposure of the cultures to 0.25% trypsin-EDTA in PBS (Thermo Fisher Scientific) and short incubation at 37 °C (~1 min). After gently tapping the plate, trypsin-EDTA with detached fibroblasts was removed and cells were further cultured in supplemented modified IMEM at 37 °C and 5% $CO_2$ until subculturing.

MCF-7 cells (RRID: CVCL_0031) and MDA-MB-453 cells (RRID: CVCL_0418) were purchased from the Global Bioresource Center (ATCC). Cells were cultivated in Dulbecco's modified eagle's medium (DMEM) supplemented with 10% FBS, 1 mM sodium pyruvate and 100 U mL$^{-1}$ penicillin and 100 µg mL$^{-1}$ streptomycin (Thermo Fisher Scientific) at 37 °C and 5% $CO_2$ in a humidified incubator.

All cells were regularly tested negative for mycoplasma contamination. Authentication of cell lines by STR was not performed. MCF-7 cells are listed as 'commonly misidentified cell lines'. Our stock of MCF-7 cells was freshly ordered from the Global Bioresource Center (ATCC). The cell morphology and behavior of all cells conformed to expectations. All experiments were performed using cells of different passages (typically passage 10–20 for MMTV-PyMT cells, and passage 18–35 for MCF-7 and MDA-MB-453 cells). The data showed high reproducibility at different passages and using different frozen aliquots from the repository or from aliquots generated in-house.

Subculturing of cells was performed when cells reached a confluency of 80–90%. Therefore, cell culture medium was removed, cells were washed 1 x with PBS, and trypsin-EDTA at a final concentration of 0.25% trypsin-EDTA in PBS was added. Subsequently, cells were incubated at 37 °C and 5% $CO_2$ in a humidified incubator until cell detachment occurred (~2–5 min). Trypsinization was stopped by adding supplemented cell culture media and cells were pelleted at 300 x $g$ for 5 min. The supernatant was removed, and cells were seeded to new cell culture dishes as required. For fluorescence microscopic live-cell imaging experiments, cells were either seeded in 6-well plates containing 1.5 H 30 mm circular glass coverslips (Paul Marienfeld GmbH, Lauda-Königshofen, Germany). All other vessels and serological pipettes used for cell culture were ordered from Corning (Kaiserslautern, Germany).

Transfection of cells was performed according to manufacturer's instructions when cells showed a confluency of ~70%, either using PolyJET DNA transfection reagent (SignaGen Laboratories, Maryland, USA) for plasmid DNA transfection or Lipofectamine 2000 (Thermo Fisher Scientific) for siRNA transfection or co-transfection of siRNA with plasmid DNA. Plasmid DNA amount was reduced to 1/3 for transfection of mitochondrial-targeted probes to ensure proper mitochondrial localization of the probes. Plasmid transfections were performed 16 hours, siRNA transfections were performed 48 hr before the experiments. Paxilline or iberiotoxin were added 12 hr prior to the experiments to the respective cell culture medium. DMSO served as a control.

## Whole-cell patch-clamp

For whole-cell patch-clamp experiments, 30,000 cells were seeded on the day before the experiment in 35 mm glass bottom μDishes (ibidi GmbH, Graefelfing, Germany) and cultivated in the respective supplemented cell culture medium over-night at 37 °C and 5% $CO_2$ in a humidified incubator. The next day, cell culture medium was removed, and cells were washed 2 x and maintained in prewarmed physiologic buffer. Subsequently, recordings were performed using borosilicate glass capillaries (0.86x1.5 × 100 mm) (Science Products GmbH, Hofheim am Taunus, Germany), with a resistance of 4–6 MW, which were pulled using a model P-1000 flaming/ brown micropipette puller (Sutter Instruments, California, USA) and filled with intracellular buffer. A MP-225 micromanipulator served for pipette control (Sutter Instruments). Recordings were performed in whole-cell mode. Currents were evoked by 15 voltage square pulses (300 ms each) from the holding potential of –70 mV to voltages between –100 mV and +180 mV delivered in 20 mV increments. For amplifier control (EPC 10) and data acquisition, Patchmaster software (HEKA Elektronik GmbH, Lambrecht, Germany) was used. Voltages were corrected offline for the capacity. Data analysis was performed using Fitmaster software (HEKA Elektronik GmbH), Nest-o-Patch software (http://sourceforge.net/projects/nestopatch, written by Dr. V Nesterov), and Microsoft Excel (Microsoft, Washington, USA).

## Cloning and plasmid preparation

Cloning was performed using conventional PCR-, restriction- and ligation-based procedures. $BK_{Ca}$-DEC was a gift from Michael J. Shipston and was N-terminally attached to an RFP ($BK_{Ca}$-DEC$^{RFP}$) using KpnI and BamHI restriction sites after PCR amplification (NEB Q5 High-Fidelity DNA-Polymerase, New England Biolabs (NEB), Ipswich, USA). For generation of $BK_{Ca}^{RFP}$, a PCR amplification of $BK_{Ca}$-DEC was performed, where the reverse primer omitted the last amino acids including the 50 amino acids encoding the DEC exon. Mitochondrial targeted TagRFP (mtRFP) was generated by fusing a double repeat of COX8 pre-sequence N-terminally to TagRFP. RFP-GPI was generated by fusing the membrane leading sequence (MLS) and the GPI-anchor signal of cadherin 13 N- and C-terminally to TagRFP, respectively. After PCR reactions, the DNA fragments were purified from gel electrophoresis using the Monarch DNA gel extraction kit (NEB), fragments and destination plasmid were digested using the respective restriction enzymes (NEB) and ligation (T4 DNA Ligase, NEB) and transformation (chemically competent NEB 5-alpha *E. coli*) were performed according to manufacturer's instructions. Plasmids were verified by sequencing (Microsynth AG, Balgach, Switzerland). DNA maxipreps were performed using the Nucleobond Xtra Maxi kit (Macherey Nagel GmbH & Co. KG, Düren, Germany). Purified DNA was stored at 4 °C.

## Confocal imaging

For confocal imaging of mtRFP, RFP-GPI, $BK_{Ca}^{RFP}$ or $BK_{Ca}$-DEC$^{RFP}$ colocalization with mitochondria, MCF-7 cells were seeded on circular 30 mm glass coverslips (Marienfeld GmbH) in six-well plates

(Corning). Cells were transfected using PolyJet transfection reagent according to the manufacturer's instructions. Sixteen hr after transfection medium was exchanged for fresh cell culture medium and cells were further cultivated for 24 hr. Subsequently, the medium was exchanged for cell equilibration buffer containing MitoGREEN (PromoCELL GmbH, Heidelberg, Germany) at a final concentration of 3 μM and cells were incubated at room temperature for 30 min. Subsequently, cells were washed 2 x with physiologic buffer, and cells were analysed using confocal fluorescence microscopy Imaging was performed using a Zeiss LSM 980 equipped with an Airyscan 2 detector. A Zeiss C Plan-Apochromat 63 x/1,4 Oil DIC M27 objective was used for all images. The ZEN 3.7 software (blue edition) was used for image acquisition and super resolution images were processed using the ZEN Airyscan module (Carl Zeiss AG). Tag-RFPs were excited at 561 nm and detected at 380–735 nm. MitoGREEN was excited at 488 nm and detected at 495–550 nm. Image analysis was performed using the colocalization test in ImageJ with Fay randomization after cell selection by ROIs.

## Fluorescence live-cell imaging

Cells were either analyzed using a Zeiss AXIO Observer Z1 or a Zeiss Axiovert 200M microscope (Carl Zeiss AG, Oberkochen, Germany). The Zeiss AXIO Observer Z1 was connected to a LEDHub high-power LED light engine (OMICRON Laserage, Rodgau-Dudenhofen, Germany) and equipped with a EC Plan-Neofluar 40 x/1.3 Oil DIC M27 objective (Carl Zeiss AG), an Optosplit II emission image splitter (Cairn Research Ltd, Faversham, UK), and a pco.panda 4.2 bi sCMOS camera (Excelitas PCO GmbH, Kelheim, Germany). The microscope possessed a BioPrecision2 automatic XY-Table (Ludl Electronic Products, Ltd., New York, USA). Optical filters included a 459/526/596 dichroic mirror and a 475/543/702 emission filter for FRET- and TMRM-based measurements, and a 409/493/573/652 dichroic mirror combined with a 514/605/730 emission filter for Dibac4(3), Fura-2 and 2-NBDG based measurements (all purchased from AHF Analysentechnik, Tübingen, Germany). The Optosplit II emission image splitter was equipped with a T505lpxr long-pass filter (AHF Analysentechnik). The LEDHub high-power LED light engine was equipped with a 340 nm, 385 nm, 455 nm, 470 nm and 505–600 nm LED, followed by the following emission filters, respectively: 340 x, 380 x, 427/10, 473/10 and 510/10 or 575/15 (AHF Analysentechnik). The Zeiss Axiovert 200M microscope was connected to a pe340$^{fura}$ light source (CoolLED, Andover, UK), an Optosplit II emission image splitter (Cairn Research Ltd.) and a pco.panda 4.2 sCMOS camera (Excelitas PCO GmbH) and equipped with 340/26, 380/14 and switchable 427/10, 485/20 or 575/15 excitation filters (AHF Analysentechnik) in the light source, respectively, a 40 x Fluar 1.30 oil immersion objective (Carl Zeiss AG), a 459/526/596 or 515LP dichroic mirror and a 475/543/702 or 525/15 emission filter (AHF Analysentechnik) in the microscope, and a T505lpxr (AHF Analysentechnik) in the Optosplit II. Image acquisition and control of both microscopes was performed using VisiView software (Visitron Systems GmbH, Puchheim, Germany). Perfusion of cells was performed using a PC30 perfusion chamber connected to a gravity-based perfusion system (NGFI GmbH, Graz, Austria) and a vacuum pump.

## Fura-2-based Ca²⁺ measurements

For fura-2-based $Ca^{2+}$ measurements, cells were taken from the humidified incubator at 37 °C and 5% $CO_2$, washed 1 x with cell equilibration buffer and loaded with fura-2 AM (Biomol GmbH, Hamburg, Germany) at a final concentration of 3.3 μM in cell equilibration buffer for 45 min at room temperature. Subsequently, cells were washed 2 x with cell equilibration buffer and stored in equilibration buffer for additional 30 min prior to the measurements. Paxilline or iberiotoxin treatment of the cells was performed 12 hr prior to the measurements and both inhibitors remained present during the fura-2 loading procedure and the measurement at concentrations of 5 μM and 30 nM, respectively. Imaging experiments were either performed on the Zeiss AXIO Observer Z1 or the Zeiss Axiovert 200M microscope (Carl Zeiss AG) in physiologic buffer using alternate excitations at 340 nm and 380 nm. Emissions were captured at roughly 514 nm (Zeiss AXIO Observer Z1) or 525 nm (Zeiss Axiovert 200M). To evoke intracellular $Ca^{2+}$ signals, cells were perfused with physiologic buffer containing ATP (Carl Roth GmbH) at a final concentration of 100 μM.

## Ca²⁺ and K⁺ calibrations

Calibrations of $[Ca^{2+}]_{cyto}$ and $[K^+]_{cyto}$ were performed by initial superfusion of cells in physiologic buffer. Subsequently, buffer was exchanged to $Ca^{2+}$ free buffer containing 5 μM of ionomycin (Alomone

Labs), and subsequent switching to 10Ca buffer for Fura-2 saturation, or 0 mM K$^+$ buffer containing 15 µM gramicidin, followed by switching to 300 mM K$^+$ buffer for saturation of NES lc-LysM GEPII 1.0. For calculation of the [Ca$^{2+}$]$_{cyto}$ and [K$^+$]$_{cyto}$ in nM and mM, respectively, the following formula was used:

$$[Ion]_{cyto} = K_d * b * \left( \frac{R - Rmin}{Rmax - R} \right)$$

## Genetically encoded sensor-based measurements

Cells were grown on 1.5 H 30 mm circular glass coverslips (Paul Marienfeld GmbH) for perfusion-based experiments. Cells were taken from the incubator, cell culture medium was removed, and cells were equilibrated for at least 30 min in cell equilibration buffer in ambient environment. Sensor plasmids used in this study comprised the following: D1ER (*Palmer et al., 2004*; Addgene plasmid #36325) for measurement of [Ca$^{2+}$]$_{ER}$ and 4mtD3cpV (*Palmer et al., 2006*) (Addgene plasmid #36324) for measurement of [Ca$^{2+}$]$_{mito}$ were a gift from Amy Palmer & Roger Tsien, mtAT1.03 (*Imamura et al., 2009*) for measurement of [ATP]$_{mito}$ was a gift from Hiromi Imamura, Laconic (*San Martín et al., 2013*) (Addgene plasmid #44238) for measurement of [LAC]$_{cyto}$ and assessment of the Warburg index was a gift from Luis Felipe Barros, mito-Hyper3 (*Bilan et al., 2013*) was a gift from Markus Waldeck-Weiermair and NES lc-LysM GEPII 1.0 (*Bischof et al., 2017*) was a gift from Roland Malli. Experiments were either performed on the AXIO Observer Z1 or the Axiovert 200M microscope (Carl Zeiss AG). For experiments where paxilline (5 µM) or iberiotoxin (30 nM) were used, these compounds were added with the media change prior to the transfection with the PolyJet transfection reagent (SignaGen Laboratories) approximately 12 hr prior to the experiments. The compounds remained present throughout the experiment. FRET-based sensors were excited at 427/10 nm and emissions were collected simultaneously at roughly 475 and 543 nm. mito-Hyper3 was excited at 427/10 and 510/10 nm. Calibration of NES lc-LysM GEPII 1.0 was performed by superfusing cells with 15 µM Gramicidin (Sigma-Aldrich Chemie GmbH) either in the absence of K$^+$ (0 mM K$^+$ buffer) or in the presence of 300 mM K$^+$ (300 mM K$^+$ buffer).

## Extracellular flux analysis

Assessment of extracellular acidification rate (ECAR) and oxygen consumption rate (OCR) was performed using a Seahorse XFe24 analyzer (Agilent, Santa Clara, USA). The day before the assay, Seahorse XF24 cell culture microplates (Agilent) were coated with 0.5 mg mL$^{-1}$ poly-L-lysine (Sigma-Aldrich Chemie GmbH) for 30 min at 37 °C, washed 2 x with PBS followed by cell seeding of 50,000 MMTV-PyMT WT, 50,000 MMTV-PyMT BK-KO or 100,000 MDA-MB-453 cells per well of the 24-well plate in a final volume of 250 µL, either in the presence or absence of 5 µM paxilline, 30 nM iberiotoxin or an equivalent amount of DMSO. 4 wells contained medium without cells and served as blank. Cells were cultivated over-night at 37 °C and 5% CO$_2$ in a humidified incubator. The Seahorse XFe24 sensor cartridges were equilibrated over-night at 37 °C in Seahorse XF Calibrant solution according to manufacturer's instructions. The next day, cells were washed using Seahorse XF DMEM medium, pH 7.4 additionally supplemented with 5.5 mM glucose, 2 mM GlutaMAX and 1 mM sodium pyruvate (Thermo Fisher Scientific), with or without paxilline, iberiotoxin or DMSO according to manufacturer's instructions. With the last washing step, a final volume of 500 µL was adjusted per well and cells were incubated at 37 °C in the absence of CO$_2$ for 30 min. Meanwhile the Seahorse XFe24 sensor cartridge was loaded with the following compounds: 55 µL of 20 µM Oligomycin-A, 62 µL of 3 µM FCCP and 69 µL of 25 µM Antimycin-A (Santa Cruz Biotechnology), yielding final concentrations of 2 µM, 300 nM and 2.5 µM upon injection, respectively. For analysis, ECAR and OCR rates were normalized for the protein concentration (µg) per well, which was assessed using the Pierce BCA protein assay kit (Thermo Fisher Scientific) according to manufacturer's instructions. Absorbance was measured at 540 nm using a TECAN multiplate reader (TECAN Group Ltd., Männedorf, Switzerland) and concentrations were assessed using a calibration curve.

## LC-MS-based metabolomics

Formic acid, acetic acid, acetonitrile and methanol of Ultra LC-MS grade were supplied by Carl Roth (Karlsruhe, Germany). Ammonium hydroxide solution (Suprapur quality 28.0–30.0% NH$_3$ basis) was purchased from Sigma-Aldrich (Merck, Taufkirchen, Germany). Deionized water was purified by a Purelab ultrapurification system (ELGA LabWater, Celle, Germany). Uniformly (U-) $^{13}$C-labeled yeast

extract of more than $2 \times 10^9$ Pichia pastoris cells (15 mg; strain CBS 7435) was obtained from ISOtopic Solutions (Vienna, Austria). All standards used were purchased from Sigma-Aldrich Chemie GmbH. Stock solutions of the individual calibrants were prepared at concentrations of 1 mg mL$^{-1}$ and used for further dilution. The individual stocks were stored at –80 °C until use.

Targeted LC-MS analysis was performed using an Agilent 1290 Infinity II series UHPLC system from Agilent Technologies (Waldbronn, Germany) equipped with a binary pump, autosampler, thermostated column compartment and a QTrap 4500 mass spectrometer with a TurboIonSpray Source from SCIEX (Ontario, Canada). The samples were filled into homogenization tubes. Internal standards were added prior to sample preparation. For the extraction of the analytes, 1 mL of the ice-cold extraction solvent (50% methanol and 50% water) and 0.15 g of zirconia/glass beads were added to the cell pellets (which were slowly thawed on ice). The samples were homogenized (6800 rpm, 1 min at 4 °C, 10×10 s, pause 30 s) with Cryolys Evolution using dry ice cooling (Bertin Technologies, France). The samples were then spun down for 5 min (16,100 x $g$ at 4 °C). The supernatant was carefully removed, transferred into fresh tubes and evaporated to dryness overnight under nitrogen using a high-performance evaporator (Genevac EZ-2) (Genevac, Ipswich, UK). The dry residue of the extract was reconstituted in 10 µL water and 90 µL acetonitrile, followed by three cycles of vortexing and sonication (30 s each). The samples were centrifuged at 18,928 x $g$ for 5 min and the supernatant was used for further analysis.

Chromatographic separation was performed on a Waters (Eschborn, Germany) Premier BEH Amide column (150x2.1 mm, 1.7 µm). For metabolite analysis in ESI$^+$ mode, mobile phases A and B were adjusted to a pH of 3.5 with formic acid and consisted of 20 mM ammonium formate in water and acetonitrile, respectively. In ESI$^-$, the chromatographic conditions differed. Mobile phase A and B were adjusted to a pH of 7.5 with acetic acid and consisted of 20 mM ammonium acetate in water and acetonitrile, respectively. The gradient elution profile was the same for both positive and negative ionization modes (0.0 min, 100% B; 13 min 70% B; 15 min 70% B; 15.1 min 100% B; 20 min 100% B) and was carried out at a flow rate of 0.25 mL min$^{-1}$ and a constant column temperature of 35 °C. The injection volume was 5 µL. The autosampler was kept at 4 °C. Ion source parameters were as follows: nebulizer gas (GS1, zero grade air) 50 psi, heater gas (GS2, zero grade air) 30 psi, curtain gas (CUR, nitrogen) 30 psi, source temperature (TEM) 450 °C, ion source voltage +5,500 V (positive mode) and –4500 V (negative mode). Due to the large number of transitions monitored simultaneously, the Scheduled-MRM function was enabled. A window of 30 s was set around the designated metabolite-specific retention time and the total cycle time was 1 s. Blank solvents (mobile phase A and B) followed by QCs were injected in the beginning of the chromatographic batch to ensure proper column and system equilibration.

## Plasma- and mitochondrial membrane potential measurements

$\Delta \Psi_{PM}$ was determined using Bis-(1,3-dibutylbarbituric acid)trimethine oxonol (Dibac4(3); Thermo Fisher Scientific). Cells were cultivated on 30 mm glass coverslips. On the day of analysis, cells were equilibrated in cell equilibration buffer containing Dibac4(3) at a concentration of 0.25 µg mL$^{-1}$ for 30 min and subsequently analyzed by fluorescence microscopy using 485/20 excitation at the Zeiss Axiovert 200M microscope. Emission was captured at roughly 525 nm.

$\Delta \Psi_{mito}$ was assessed using Tetramethylrhodamin-Methylester (TMRM) (Thermo Fisher Scientific). After cell cultivation on 30 mm glass coverslips in the presence or absence of paxilline, iberiotoxin or DMSO for 12 hr, the cell culture medium was replaced with cell equilibration buffer containing 200 nM TMRM and cells were incubated in ambient environment for 30 minutes. Subsequently, cells were washed with physiologic buffer containing 2 mM or 25 mM glucose and 200 nM TMRM, and cells were equilibrated for further 30 min. The glass coverslips were transferred to the PC30 perfusion chamber and experiments were started using the gravity-based perfusion system (NGFI GmbH). Paxilline, iberiotoxin or DMSO (control) and 200 nM TMRM remained present throughout the experiment. Mitochondrial depolarization was induced by the perfusion of a buffer containing 200 nM FCCP. For analysis, the fluorescence emission at ≥600 nm upon excitation at 575/15 nm of a region of interest (ROI) above mitochondria was divided by a ROI in the nucleus (mitochondria free area) and the ratio was plotted over-time, or basal values were given.

## 2-NBDG based glucose uptake measurements

Glucose uptake was assessed using 2-(N-(7-Nitrobenz-2-oxa-1,3-diazol-4-yl)Amino)–2-Deoxyglucose (2-NBDG) (Biomol GmbH). Therefore, cells were seeded on 30 mm glass coverslips (Marienfeld GmbH)

in 6-well plates (Corning) and cultivated over-night at 37 °C and 5% $CO_2$ in a humidified incubator. The next day, cells were taken from the incubator, cell culture medium was replaced for cell equilibration buffer and cells were equilibrated for 30 min at ambient environment. Subsequently, cells were washed 3 x with glucose-free physiologic buffer, and glucose free physiologic buffer containing 100 µM 2-NBDG, with or without paxilline and 200 nM FCCP or an equivalent amount of DMSO, was added to the cells, followed by incubation at 37 °C for 30 min. Next, cells were washed 3 x with glucose free physiologic buffer to remove any residual 2-NBDG and were analyzed by fluorescence microscopy using 485/20 excitation light. Emission was captured at roughly 525 nm (Zeiss Axiovert 200M).

## Mitochondrial single-channel patch-clamp measurements

For mitoplast electrophysiology, mitochondria were isolated from BCC grown to confluency (>90%). Adherent cells were washed 2 x with PBS, scraped from the dish, collected in a tube and centrifuged at 400 x $g$ for 5 min. Subsequently, the cell pellet was resuspended in preparation solution, followed by homogenization using a glass-glass homogenizer. Next, the homogenate was centrifuged at 9200 x $g$ for 10 min. The resulting pellet was resuspended in preparation solution and centrifuged at 780 x $g$ for 10 min. The supernatant was collected, followed by centrifugation at 9200 x $g$ for 10 min and resuspension of the mitochondrial fraction in the mitochondrial storage buffer. All procedures were performed at 4 °C.

An osmotic swelling procedure was used for the preparation of mitoplasts from the isolated mitochondria. Therefore, mitochondria were added to the hypotonic buffer for ~1 min to induce swelling and breakage of the outer mitochondrial membrane. Subsequently, isotonicity of the solution was restored by the addition of hypertonic buffer at a dilution of 1:5.

Patch-clamp experiments on mitoplasts were performed as previously described (*Bednarczyk et al., 2013*). The experiments were carried out in mitoplast-attached single-channel mode using borosilicate glass pipettes with a mean resistance of 10–15 MΩ. The patch-clamp glass pipette was filled with mitochondrial storage buffer. This isotonic solution was used as a control solution for all experiments. The size of the pipettes and the formation of the gigaseal were monitored by measuring electrical resistance. Connections were made with Ag/AgCl electrodes and an agar salt bridge (3 M KCl) for the ground electrode. The current was recorded using an Axopatch 200B patch-clamp amplifier (Molecular Devices, California, USA). To apply substances, a self-made perfusion system containing a holder with a glass pipe, a peristaltic pump, and Teflon tubing was used. All channel modulators were added as dilutions in the isotonic solution containing 100 µM $CaCl_2$.

The presented single-channel current-time recordings are representative for the most frequently observed conductances under the given conditions. The conductance was calculated from the current-voltage relationship. The probability of channel openings and the current amplitude were determined using the Single-Channel Search mode of the Clampfit 10.7 software (Molecular Devices).

## Western blotting

Mitochondria for western blotting were prepared as described earlier (*Pallotti and Lenaz, 2007*) with some modifications. Cells were washed and collected in PBS and centrifuged at 500 x $g$ for 10 min. The pellet was frozen in liquid nitrogen and stored at –80 °C. The next day, pellet was resuspended in ice-cold isolation buffer containing (in mM): 210 mannitol, 70 sucrose, 1 PMSF, 5 HEPES and bovine serum albumin (2.5 mg mL$^{-1}$), pH 7.2. Digitonin was added to a final concentration of 0.02–0.04% for additional membrane permeabilization. After 1 min of incubation, digitonin was diluted with isolation buffer and the cells were centrifuged at 3000 x $g$ for 5 min at 4 °C. The pellet was resuspended in ice-cold isolation buffer. The cells were homogenized using a glass/glass homogenizer and centrifuged at 1000 x $g$ for 5 min at 4 °C followed by another homogenization. The homogenates were centrifuged at 1000 x $g$ for 5 min at 4 °C to remove cell remnants. The supernatants were collected and centrifuged for 60 min at 10,000 x $g$, at 4 °C. Next, the pellet was resuspended in the isolation buffer without BSA and centrifuged at 10,000 x $g$ for 30 min at 4 °C, followed by resuspension in isolation buffer without BSA and centrifugation at 10,000 x $g$ for 15 min at 4 °C. This step was repeated once more. The pellet containing crude mitochondria was resuspended in 0.8 mL of 15% Percoll (in isolation buffer without BSA) and layered on top of a Percoll step gradient (23% and 40% Percoll layers). The suspension was centrifuged at 30,000 x $g$ for 30 min at 4 °C. Mitochondrial fraction located between the 23% and 40%

Percoll layer was collected, diluted with isolation buffer without BSA and centrifuged at 10,000 x $g$ for 15 min at 4 °C. The final mitochondrial pellet was resuspended in storage buffer containing 500 mM sucrose and 5 mM HEPES, pH 7.2. Each mitochondrial isolation was performed using between 15 and 30 x $10^6$ cells.

A given amount of sample solubilized in Laemmli buffer (Bio-Rad) was separated by 10% Tris-tricine-SDS-PAGE and transferred onto polyvinylidene difluoride (PVDF) membranes (Bio-Rad). After protein transfer, the membranes were blocked with 10% nonfat dry milk solution in Tris-buffered saline with 0.2% Tween 20 and exposed to one of the following antibodies: anti-BKα antibody (NeuroMabs, USA, clone L6/60, diluted 1:200), anti-COXIV (Cell Signaling Technology, Leiden, Netherlands, no. 4844, 1:1000), or anti-alpha 1 sodium potassium ATPase (Abcam, Berlin, Germany, no. ab7671, 1:1000). This was followed by incubation with a secondary anti-rabbit (Thermo Fisher Scientific, no. 31460) or anti-mouse antibody (Thermo Fisher Scientific, no. SA1-100) coupled to horseradish peroxidase. The blots were developed using enhanced chemiluminescence solution (GE Healthcare). To estimate the molecular weight of the analyzed proteins PageRuler Prestained Protein Ladder (Thermo Fisher Scientific) was used.

## qPCR analysis

mRNA was isolated from six-well plates showing a cell confluency of ~90% using the Monarch Total RNA Miniprep kit (NEB). siRNA transfection was performed 48 hr prior to RNA isolation using the Lipofectamine 2000 transfection reagent (Thermo Fisher Scientific) according to manufacturer's instructions. qPCR reactions were performed using the GoTaq 1-Step RT-qPCR System (Promega GmbH, Walldorf, Germany). A total of 100 ng of isolated mRNA were used per reaction. Primers were designed to span exon – exon junctions and to recognize both, human and murine $BK_{Ca}$ sequences. Different primer pairs were used for human and murine b-tubulin. Primer and siRNA sequences are listed in *Supplementary file 1a* and *Supplementary file 1b*. Primers were purchased from Thermo Fisher Scientific, siRNAs were purchased from Microsynth AG. qPCR reactions were run in a CFX connect real-time PCR instrument (Bio-Rad Laboratories GmbH, Feldkirchen, Germany). Sample (Ct) values were normalized to Ct's of housekeeper gene (b-Tubulin) and calculated as $2^{-Ct (normalized)}$.

## Proliferation assays

Proliferation assays were performed using 3-(4,5-dimethylthiazol-2-yl)–2,5-diphenyltetrazolium bromide (MTT) (Thermo Fisher Scientific) assay. For the assay, 2.500 MMTV-PyMT WT and BK-KO cells or 7.500 MDA-MB-453 cells were seeded per well in flat bottom 96-well cell culture plates (Corning) in their respective cell culture medium. Per run, condition, and time-point, seven technical replicates were performed. One well served as a blank per time-point and condition and did not contain cells. The next day, medium was exchanged for fresh cell culture medium either containing 5 µM paxilline, 30 nM iberiotoxin or an equivalent amount of DMSO as a control and the MTT assay was started for time-point 24 hr directly thereafter. For each time-point (24, 48, 72, and 96 hr), MTT reagent was added with a medium change (10 µL+100 µL / well) at a final concentration of 455 µg mL$^{-1}$ and cells were incubated for 4 hr. Subsequently, 85 µL of medium were removed from each well, 85 µL of DMSO (Carl Roth GmbH) was added and wells were mixed by pipetting thoroughly. Of this mixture, 85 µL were transferred to a new 96-well plate (Corning) and absorbance was measured at 540 nm using a TECAN microplate reader (TECAN Group Ltd.). Absorbances at the different time-points (48, 72, and 96 hr) were normalized to the absorbances after 24 hr.

For KI-67-based assessment of cell proliferation rates, 20,000 MMTV-PyMT WT and BK-KO cells and 30,000 MDA-MB-453 cells were seeded in µ-slide eight wells (ibidi GmbH). The next day, cells were washed 1 x with PBS, and serum-free cultivation medium was added. Cells were further cultivated for 48 hr, prior to addition of DMSO (ctrl) 5 µM paxilline or 30 nM iberiotoxin. After additional 24 hr, serum containing medium supplemented with DMSO, paxilline, or iberiotoxin was re-added, and cell growth was allowed for another 48 hr. Finally, cells were washed 2 x with PBS and fixed with ice cold methanol/aceton (50:50 v/v) for 15 min at –20 °C. Subsequently, cells were washed thrice with PBS, and incubated in PBS containing 5% bovine serum albumin (BSA, Carl Roth GmbH) for 1 hr. Subsequently, anti Ki-67 rabbit monoclonal antibody (D3B5, Cell Signaling Technology, RRID: AB_2687446) in 5% BSA-PBS was added at a dilution of 1:500. Wells without primary antibody served as a negative control. Slides were incubated over-night at 4 °C. The next day, antibody solution was removed,

cells were washed 3 x with PBS, and subsequently incubated with Goat anti-Rabbit IgG (H+L) Cross-Adsorbed Secondary Antibody, Alexa Fluor 488 (Thermo Fisher Scientific, RRID: AB_2633280) for 2 hr at room temperature, followed by washing 2 x with PBS. During a 3rd washing step, Hoechst 33342 (Thermo Fisher Scientific) at a dilution of 1:2000 was added, cells were incubated for 10 min at room temperature and washed an additional time with PBS. Ultimately, cells were mounted in PermaFluor Mountant (Microm International GmbH, Dreieich, Germany), kept over-night at 4 °C, and imaged using the Zeiss Axio Observer Z1 wide-field microscope. AlexaFluor 488 and Hoechst 33342 were illuminated using 473/10 and 380 x filters, respectively. Two images were captured per replicate. Image analysis was performed using ImageJ. Threshold positive area of Ki-67 (AlexaFluor488) was expressed as fraction of the threshold positive area of Hoechst 33342 per image, and the average of the two images per well was calculated.

## Colony formation assays

For colony formation assays, cell suspensions containing 400 cells mL$^{-1}$ of MMTV-PyMT WT or BK-KO cells were prepared. 1 mL of this suspension was spread drop-per-drop per well of a 6-well cell culture plate (Corning) already containing 1 mL of cell culture medium using 1 mL syringes and 23 G needles. Culture plates were gently moved left/right and back/forth to equally distribute the cells across the well, plates were kept at room temperature for 20 min to ensure uniform cell settling, and cells were subsequently cultured over-night at 37 °C and 5% $CO_2$ in a humidified incubator. The next day, medium was exchanged for 2 mL of fresh cell culture medium, and cells were kept in the presence or absence of $O_2$ at 37 °C and 5% $CO_2$ in humidified incubators for 7 days. After 7 days, cells were washed carefully 2 x with PBS and fixed for 30 min on ice using 2% PFA in PBS. Meanwhile, a solution of crystal violet (0.01% w/v) was prepared in dd$H_2O$, cells were washed 2 x with PBS and incubated in the crystal violet solution (Sigma-Aldrich Chemie GmbH) for 60 mins after fixation, followed by excessive washing with dd$H_2O$ until a clear background was obtained. Plates were dried at room temperature for at least 12 hr before images were captured using an Amersham Imager 600 (GE Healthcare UK Limited, Buckinghamshire, UK). Analysis was performed using ImageJ.

## Breast cancer patient study

A subset of 551 primary tumor specimens with available archival tumor blocks were obtained from a prospective, observational multicenter adjuvant endocrine treatment study of 1286 post-menopausal HR-positive breast cancer patients recruited between 2005 and 2011 (German Clinical Trial Register DRKS 00000605, 'IKP211' study). Study inclusion and exclusion criteria have been previously described (*Schroth et al., 2020*). Patients had received standard endocrine treatment, that is tamoxifen, aromatase inhibitors, or switch regimens between both. Ethics approval was obtained from the Medical Faculty of the University of Tübingen and the local ethics committees of all participating centers in Germany. Informed patient consent was obtained from all participants as required by institutional review boards and research ethics committees. All patient data were de-identified prior inclusion in this study.

## Nanostring nCounter gene expression analysis

Gene expression analysis was performed according to manufacturer's protocol. In brief, total RNA from human primary tumors of the IKP211 study (Quick-DNA/RNA FFPE, ZymoResearch, Freiburg, Germany) was extracted from 10 µm sections of FFPE tissues with at least 20% tumor cell content. Samples were enriched for tumor tissue by microdissection. A total of 250 ng RNA was subsequently hybridized with target-specific capture and color-coded reporter probe sets in a pre-warmed thermal cycler at 65 °C for 20 hr. In the post-hybridization process, the total volume was increased to 32 µl with RNase-free water. Fluorescence count measurements were immediately conducted in the Nanostring nCounter System. Data were analyzed using nSolver 4.0 and normalized to housekeeping genes. ABCF1, NRDE2, POLR2A, PUM1, and SF3A1 served as housekeepers. Probes used for Nanostring nCounter gene expression analysis are listed in *Supplementary file 1c*.

## Statistical analysis

Statistical analysis was performed using Prism8 software (GraphPad Software, Boston, USA). All data were tested for normal distribution using D'Agostino and Pearson omnibus normality test. A two-tailed

Unpaired t-test or Welch's t-test were used for statistical comparison of normally distributed data, depending on whether the variances of the populations were comparable or significantly different as tested by an F-test. A two-tailed Mann-Whitney test was used for pairwise comparison of non-normally distributed data. Comparison of >2 data sets was done either using One-way ANOVA followed by Tukey's multiple comparison (MC) test or Brown-Forsythe and Welch ANOVA test followed by Games-Howell's MC test for normally distributed data, depending on whether the variances were comparable or significantly different. A Kruskal-Wallis test followed by Dunn's MC test was performed if data were not normally distributes. The statistical tests used are indicated in the figure legends. p-values of ≤0.05 were considered as significant, where * or #p≤0.05, ** or †p≤0.01 and *** or ‡p≤0.001. No priory sample size estimation was performed.

## Material availability

Materials are available from the corresponding author upon request.

## Acknowledgements

This work was funded by the Deutsche Forschungsgemeinschaft (DFG) with individual grants to RL with LM, and under Germany's Excellence Strategy-EXC 2180–390900677 (MS). RL is a member of the GRK2381: "cGMP: From Bedside to Bench", DFG grant number 335549539. SM, WS, MS and RL acknowledge financial support from the ICEPHA Graduate Program "Membrane-associated Drug Targets in Personalized Cancer Medicine". HB is a fellow of the Austrian Science Fund (FWF) funded Erwin-Schrödinger-Program, project number J-4457. SB acknowledges financial support from the Fritz Thyssen Stiftung. The authors thank Clement Kabagema-Bilan, Michael Glaser and Antoni Wrzosek for excellent technical support and Peter Ruth for valuable discussions. This work was partially supported by the Nencki Institute of Experimental Biology, the Polish National Science Centre grant no. 2019/34/A/NZ1/00352 (AS), and by the FWF, project number I-3716 (RM). WS, FB and MS are supported in part by the Robert Bosch Stiftung, Stuttgart, Germany.

## Additional information

### Funding

| Funder | Grant reference number | Author |
|---|---|---|
| Deutsche Forschungsgemeinschaft | 335549539 | Robert Lukowski |
| Interfaculty Centre for Pharmacogenomics and Pharma Research | Membrane associated Drug Targets in Personalized Cancer Medicine | Selina Maier Werner Schroth Matthias Schwab Robert Lukowski |
| Austrian Science Fund | J-4457 | Helmut Bischof |
| Fritz Thyssen Stiftung | | Sandra Burgstaller |
| Narodowe Centrum Nauki | 2019/34/A/NZ1/00352 | Adam Szewczyk |
| Austrian Science Fund | I-3716 | Roland Malli |
| Deutsche Forschungsgemeinschaft | LU 1490/12-1 | Robert Lukowski |
| Deutsche Forschungsgemeinschaft | Germany's Excellence Strategy-EXC 2180–390900677 | Matthias Schwab |
| Deutsche Forschungsgemeinschaft | LU 1490/10-1 | Robert Lukowski |
| Deutsche Forschungsgemeinschaft | MA 8113/2-1 | Lucas Matt |

| Funder | Grant reference number | Author |
| --- | --- | --- |
| Robert Bosch Stiftung | | Werner Schroth<br>Florian A Büttner<br>Matthias Schwab |

The funders had no role in study design, data collection and interpretation, or the decision to submit the work for publication.

## Author contributions

Helmut Bischof, Conceptualization, Data curation, Formal analysis, Funding acquisition, Investigation, Visualization, Methodology, Writing – original draft; Selina Maier, Piotr Koprowski, Bogusz Kulawiak, Sandra Burgstaller, Data curation, Formal analysis, Investigation, Writing – review and editing; Joanna Jasińska, Monika Zochowska, Data curation, Investigation; Kristian Serafimov, Data curation, Formal analysis, Investigation; Dominic Gross, Lucas Matt, David Arturo Juarez Lopez, Methodology; Werner Schroth, Data curation, Formal analysis, Funding acquisition, Investigation, Methodology, Writing – review and editing; Ying Zhang, Data curation, Methodology; Irina Bonzheim, Resources, Data curation, Investigation; Florian A Büttner, Data curation, Formal analysis, Investigation, Methodology; Falko Fend, Andreas L Birkenfeld, Roland Malli, Resources, Methodology; Matthias Schwab, Resources, Funding acquisition, Methodology, Writing – review and editing; Michael Lämmerhofer, Resources, Validation, Methodology; Piotr Bednarczyk, Data curation, Formal analysis, Investigation, Visualization, Methodology, Writing – review and editing; Adam Szewczyk, Resources, Supervision, Funding acquisition, Methodology, Writing – review and editing; Robert Lukowski, Conceptualization, Resources, Supervision, Funding acquisition, Validation, Methodology, Writing – original draft, Project administration

## Author ORCIDs

Helmut Bischof ⓘ http://orcid.org/0000-0003-2380-600X
Bogusz Kulawiak ⓘ http://orcid.org/0000-0002-4420-9557
Sandra Burgstaller ⓘ http://orcid.org/0000-0002-3926-7023
Irina Bonzheim ⓘ http://orcid.org/0000-0002-7732-0788
Andreas L Birkenfeld ⓘ http://orcid.org/0000-0003-1407-9023
Roland Malli ⓘ http://orcid.org/0000-0001-6327-8729
Piotr Bednarczyk ⓘ http://orcid.org/0000-0002-7125-0715
Adam Szewczyk ⓘ http://orcid.org/0000-0001-5519-260X
Robert Lukowski ⓘ http://orcid.org/0000-0002-4564-3574

## Ethics

Tumor growth in vivo and biopsies were authorized by the local ethics Committee for Animal Research (Regierungspräsidium Tübingen (PZ1/16, PZ2/17, PZ4/20 G)), and were performed in accordance with the German Animal Welfare Act.

Reviewer #1 (Public Review): https://doi.org/10.7554/eLife.92511.3.sa1
Reviewer #2 (Public Review): https://doi.org/10.7554/eLife.92511.3.sa2
Reviewer #3 (Public Review): https://doi.org/10.7554/eLife.92511.3.sa3
Author response https://doi.org/10.7554/eLife.92511.3.sa4

# Additional files

## Supplementary files

• Supplementary file 1. File contains the Supplementary Tables for the Manuscript. (a) Probes used for Nanostring nCounter gene expression analysis. (b) Primers used for qPCR analysis. (c) siRNAs used for silencing based experiments.

• MDAR checklist

## Data availability

Source data files have been provided for all figures and panels.

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
