## [Editor Report · eLife assessment]

The large-conductance Ca^2+^ activated K^+^ channel BK_Ca_ has been reported to promote breast cancer progression. The present study presents **convincing** evidence that an intracellular subpopulation of this channel reprograms breast cancer cells towards the Warburg phenotype, one of the metabolic hallmarks of cancer. This **important** finding advances the field of cancer cell metabolism and has potential therapeutic implications.

---

## [Referee Report · Reviewer #1 (Public Review)]

Original Review:

Bischoff et al present a carefully prepared study on a very interesting and relevant topic: the role of ion channels (here a Ca2+-activated K+ channel BK) in regulating mitochondrial metabolism in breast cancer cells. The potential impact of these and similar observations made in other tumor entities has only begun to be appreciated. That being said, the authors pursue in my view an innovative approach to understanding breast cancer cell metabolism.

Considering the following points would further strengthen the manuscript:

Methods:

(1) The authors use an extracellular Ca2+ concentration (2 mM) in their Ringer's solutions that is almost twice as high as the physiologically free Ca2+ concentration (ln 473). Moreover, the free Ca2+ concentration of their pipette solution is not indicated (ln 487).

(2) Ca2+I measurements: The authors use ATP to elicit intracellular Ca2+ signals. Is this then physiological stimulus for Ca2+ signaling in breast cancer? What is the rationale for using ATP? Moreover, it would be nice to see calibrated baseline values of Ca2+i

(3) Membrane potential measurements: It would be nice to see a calibration of the potential measurements; this would allow to correlate IV relationship with membrane potential. Without calibration it is hard to compare unless the identical uptake of the dye is shown.

Do paxilline or IbTx also induce a depolarization?

(4) mito-potential measurements: Why did the authors use such a long time course and preincubated cells mit channel blockers overnight? Why did they not perform paired experiments and record the immediate effect of the BK channel blockers in the mito potential?

(5) MTT assay are also based on mitochondrial function - since modulation of mito function is at the core of this manuscript, an alternative method should be used.

Results:

(1) Fig. 5G: The number of BK "positive" mitoplasts is surprisingly low - how does this affect the interpretation? Did the authors attempt to record mitoBK current in the "whole-mitoplast" mode? How does the mitoBK current density compare with that of the plasma membrane? Is it possible to theoretically predict the number of mitoBK channels per mitochondrium to elicit the observed effects? Can these results be correlated with immuno-localization of mitoBK channels?

(2) There are also reports about other mitoK channels (e.g. Kv1.3, KCa3.1, KATP) playing an important role in mitochondrial function. Did the authors observe them, too? Can the authors speculate on the relative importance of the different channels? Is it known whether they are expressed organ-/tumor-specifically?

Comments on revised version:

The authors responded to all of my comments - except for one - in a satisfactory way so that I have no further concerns. The authors have prepared a very interesting piece of work that advances the field.

However, I disagree with respect to their interpretation of statistics. Individually analyzed cells are not the best biological replicate per se. In my view a true replicate requires the use of an independent batch of cells derived from a new passage. The statistical analysis can only based on the total number of n cells, if each replicate contributes the same number of cells. If this is not the case, the authors will have to calculate the average of each replicate first so that they are equally weighted.

---

## [Referee Report · Reviewer #2 (Public Review)]

Summary:

The large-conductance Ca2+ activated K+ channel (BK) has been reported to promote breast cancer progression, but it is not clear how. The present study, carried out in breast cancer cell lines, concludes that BK located in mitochondria reprograms cells towards the Warburg phenotype, one of the metabolic hallmarks of cancer.

Strengths:

The use of a wide array of modern complementary techniques, including metabolic imaging, respirometry, metabolomics and electrophysiology. On the whole experiments are astute and well designed, and appear carefully done. The use of a BK knock out cells to control for the specificity of the pharmacological tools is a major strength. The manuscript is clearly written. There are many interesting original observations that may give birth to new studies.

Weaknesses: The main conclusion regarding the role of a BK channel located in mitochondria appears is not sufficiently supported. Other perfectible aspects are the interpretation of co-localization experiments and the calibration of Ca2+ dyes. These points are discussed in more detail in the following paragraphs:

(1) May the metabolic effects be ascribed to a BK located in mitochondria? Unfortunately not, at least with the available evidence. While it is clear these cells have a BK in mitochondria (characteristic K+ currents detected in mitoplasts) and it is also well substantiated that the metabolic effects in intact cells are explained by an intracellular BK (paxilline effects absent in the BK KO), it does not follow that both observations are linked. Given that ectopic BK-DEC appeared at the surface, a confounding factor is the likely expression of BK in other intracellular locations such as ER, Golgi, endosomes, etc. To their credit authors acknowledge this limitation several times throughout the text ("...presumably mitoBK...") but not in other important places, particularly in title and abstract.

(2) mitoBK subcellular location. Pearson correlations of 0.6 and about zero were obtained between the locations of mitoGREEN on one side, and mRFP or RFP-GPI on the other (Figs. 1G and S1E). These are nice positive and negative controls. For BK-DECRFP however the Pearson correlation was about 0.2. What is the Z resolution of apotome imaging? Assuming an optimum optical section of 600 nm, as obtained a 1.4 NA objective with a confocal, that mitochondria are typically 100 nm in diameter and that BK-DECRFP appears to stain more structures that mitoGREEN, the positive correlation of 0.2 may not reflect colocalization. For instance, it could be that BK-DECRFP in not just in mitochondria but in a close underlying organelle e.g. the ER. Along the same line, why did BK-RFP also give a positive Pearson? Isn´t that unexpected? Considering that BK-DEC was found by patch clamping at the plasma membrane, the subcellular targeting of the channel is suspect. Could it be that the endogenous BK-DEC does actually reside exclusively in mitochondria (a true mitoBK), but overflows to other membranes upon overexpression? Regarding immunodetection of BK in the mitochondrial Percoll preparation (Fig. S5), absence of NKA demonstrates absence of plasma membrane contamination, but does not inform about contamination by other intracellular membranes.

(3) Calibration of fluorescent probes. The conclusion that BK blockers or BK expression affects resting Ca2+ levels should be better supported. Fluorescent sensors and dyes provide signals or ratios that need be calibrated if comparisons between different cell types or experimental conditions are to be made. This is implicitly acknowledged here when monitoring ER Ca2+, with an elaborate protocol to deplete the organelle in order to achieve a reading at zero Ca2+.

(4) Line 203. "...solely by the expression of BKCa-DECRFP in MCF-7 cells". Granted, the effect of BKCa-DECRFP on the basal FRET ratio appears stronger than that of BK-RFP, but it appears that the latter had some effect. Please provide the statistics of the latter against the control group (after calibration, see above).

The revised version of the manuscript has incorporated my suggestions to a very reasonable degree, in several cases with new experiments. The details of these improvements can be found in the correspondence.

---

## [Referee Report · Reviewer #3 (Public Review)]

The original research article, titled "mitoBKCa is functionally expressed in murine and human breast cancer cells and promotes metabolic reprogramming" by Bischof et al, has demonstrated the underlying molecular mechanisms of alterations in the function of Ca2+ activated K+ channel of large conductance (BKCa) in the development and progression of breast cancer. The authors also proposed that targeting mitoBKCa in combination with established anti-cancer approaches, could be considered as a novel treatment strategy in breast cancer treatment.

The paper is modified according to the reviewer's comments. Most of the queries raised by this reviewer were answered. However, the preclinical implication of this study can also be manifested in combinatorial treatment with known chemotherapeutic drugs which is lacking in this manuscript. Hopefully, the authors will consider this in their future study.

---

## [Author Response]

The following is the authors’ response to the current reviews.

Our answer to the final point(s) raised is as follows:

"We thank the reviewer for the comment. We checked our datasets accordingly. Typically, the n of cells showed deviations of maximally 20% from experiment to experiment (e.g. 16-24 cells per experiment). Additionally, experiments were performed using different passages of the cells. Moreover, data were validated at different time-points during the study using newly thawed cell lines."

The following is the authors’ response to the original reviews.

**Public Reviews:**

**Reviewer #1 (Public Review):**
Bischoff et al present a carefully prepared study on a very interesting and relevant topic: the role of ion channels (here a Ca2+-activated K+ channel BK) in regulating mitochondrial metabolism in breast cancer cells. The potential impact of these and similar observations made in other tumor entities has only begun to be appreciated. That being said, the authors pursue in my view an innovative approach to understanding breast cancer cell metabolism. Considering the following points would further strengthen the manuscript:

We thank reviewer #1 for the overall positive feedback on our study.

Methods:(1) The authors use an extracellular Ca2+ concentration (2 mM) in their Ringer's solutions that is almost twice as high as the physiologically free Ca2+ concentration (ln 473). Moreover, the free Ca2+ concentration of their pipette solution is not indicated (ln 487).

Indeed, we utilized 2 mM of Ca2+ in the physiologic live-cell imaging buffer. This concentration could actually be a little lower than the total Ca2+ concentration (ranging usually from 2.2 to 2.6 mM) in the body, while the free Ca2+ concentration is typically half as high. Nevertheless, we find multiple studies different from ours, which utilized 2 mM for their live-cell-based experiments. Please check the following studies, which represent only a small selection:

https://doi.org/10.1038/s41598-019-49070-8

https://doi.org/10.1016/j.bpj.2020.08.045

https://doi.org/10.1016/j.redox.2022.102319

However, to ensure that the applied conditions are physiologically relevant, we reperformed experiments using MMTV-PyMT WT and MMTV-PyMT BK-KO cells and compared cytosolic Ca2+ concentrations over time in response to cell stimulation with ATP, either in the presence of 1.0 mM (Author response image 1A) or 2.0 mM extracellular Ca2+ (Author response image1B). The respective graphs are attached in the following for reviewer’s inspection. As expected, we find that the intracellular Ca2+ concentration in MMTV-PyMT WT and BK-KO cells was dependent on the extracellular Ca2+ concentration. Importantly, however, irrespective of the exact Ca2+ concentration applied, we observed a similar difference in basal cytosolic Ca2+ between MMTV-PyMT WT and BK-KO cells (Author response image1C).

**Author response image 1. sa4fig1:** Cytosolic Ca2+ concentrations over-time in the presence of 1. mM or 2.0 mM extracellular Ca2+.

Concerning the Ca2+ concentration in the patch-pipette – we are very glad that you uncovered an error in our description and apologize for the mistake. Actually, the information the reviewer is referring to was already given in the previous version of the manuscript, but unclear because a comma was shifted (see line 487 in the originally submitted manuscript). The Ca2+ concentration of the patch-pipette was 0.1 mM in the presence of 0.6 mM EGTA, which should (according to Ca-EGTA calculator, https://somapp.ucdmc.ucdavis.edu/pharmacology/bers/maxchelator/CaEGTA-NIST.htm) be equivalent to ~30 nM of free Ca2+ in the patch pipette. We corrected the mistake in the manuscript and thank the reviewer again for spotting this inaccuracy.

(2) Ca2+I measurements: The authors use ATP to elicit intracellular Ca2+ signals. Is this then a physiological stimulus for Ca2+ signaling in breast cancer? What is the rationale for using ATP? Moreover, it would be nice to see calibrated baseline values of Ca2+i.

We thank the reviewer for the comment and suggestion. Importantly, it was demonstrated recently, that all of the utilized cell lines respond to treatment with extracellular ATP with a prominent increase in Ca2+I, most probably indicating the expression of purinergic receptors, which was a prerequisite to observe ATP induced changes in [Ca2+]i.

https://doi.org/10.1038/s41419-022-05329-z,

https://doi.org/10.1093/carcin/bgt493

https://doi.org/10.1038/s41598-018-26459-5

Furthermore, ATP plays a crucial role in the tumor microenvironment, where high rates of cell death occur. Hence, ATP is of pathophysiologic relevance for the utilized cancer cell lines.

https://doi.org/10.1038/s41568-018-0037-0

https://doi.org/10.3390/cells9112496

https://doi.org/10.1002/jcp.30580

Following the suggestions by Reviewer #1 (and #2), we included calibrations of Ca2+cyto and Ca2+mito in the manuscript, by depleting the intracellular Ca2+ stores using Ionomycin in the absence of extracellular Ca2+ (EGTA) to validate the basal difference in Ca2+cyto and Ca2+mito. Additionally, Ca2+cyto was calibrated under basal and inhibitor treated conditions, and values in nM are given in the text (p. 5, lines 185-190, 193-195 and 199-200, in the tracked changes version of the MS). The new data can be found in new Figure S2F – Figure S2J and new Figure S2R – Figure S2V. Moreover, we calculated basal [Ca2+]cyto in the different BKCa pro- and deficient cell lines and under inhibitor treated conditions. We additionally added information about the pathophysiologic relevance of ATP in the tumor microenvironment in lines 175-178 in the tracked changes version of the manuscript.

(3) Membrane potential measurements: It would be nice to see a calibration of the potential measurements; this would allow us to correlate the IV relationship with membrane potential. Without calibration, it is hard to compare unless the identical uptake of the dye is shown. Does paxilline or IbTx also induce depolarization?

We thank the reviewer for the suggestion. Indeed, membrane potential calibrations/ measurements using the membrane potential sensitive dye Dibac4(3) would be interesting, however, technically hardly feasible. The reason is that the principle of the dye is based on different uptake in response to differences in membrane potential, and not ratiometric as for most other dyes/ sensors used. Considering this limitation, we decided to perform membrane potential measurements by patch-clamp analysis. Additionally, we performed these experiments upon inhibition of PM-located BKCa by IBTX. Current-clamp experiments confirmed the difference in basal membrane potential between MMTV-PyMT WT and BK-KO cells (consult new Figure S1C and lines 127-130 in the tracked changes version of the manuscript). Interestingly, IBTX treatment depolarized the PM potential to the BK-KO cell level, which validates that BK activity and PM potential are connected. In addition to this approach, we utilized our recently developed genetically encoded K+ sensors revealing basal differences in [K+]cyto between MMTV-PyMT WT and BK-KO cells. Also this difference between both genotypes was equalized by IBTX as the respective treatment increased [K+]cyto only in WT cells, which most likely explains the cause of PM depolarization (consult lines 130-135 in the tracked changes version of the manuscript and new Figure S1D and Figure S1E).

(4) Mito-potential measurements: Why did the authors use such a long time course and preincubate cells with channel blockers overnight? Why did they not perform paired experiments and record the immediate effect of the BK channel blockers in the mito potential?

We thank the reviewer for the suggestion. We performed TMRM-based experiments with MMTV-PyMT WT cells in response to short-term exposure to paxilline, which did not significantly affect the mitochondrial membrane potential, at least within 15 minutes of treatment (Author response image 2). This indicates, that further downstream processes subsequent to (mito) BKCa inhibition affect the mitochondrial membrane potential(MMP), most probably including remodeling processes of the respiratory chain, mitochondrial ion homeostasis or glycolytic activity, ultimately also delivering reduction equivalents to mitochondria. Our final goal was to validate potential differences between a BKCa pro-and deficient cell model, whereby the latter cells lacked the BKCa channel since its origination. Hence, “long-term” (~12h) BKCa inhibition as performed in our experiments rather reflects the BK-KO cell situation. Taken together with the new experiment (Author response image 2), we can now state that the effect of BK inhibition on the MMP is at least not the consequence of an acute (within minutes) channel blockade.

**Author response image 2. sa4fig2:** Mitochondrial membrane potential, as measured using TMRM, in response to acute short-term administration of 5µM paxilline, followed by mitochondrial depolarization using FCCP.

(5) MTT assays are also based on mitochondrial function - since modulation of mito function is at the core of this manuscript, an alternative method should be used.

We thank the reviewer for the important comment. We performed additional, immunofluorescence-based experiments using Ki-67 staining to assess cell proliferation rates. The newly added data can be found in the text, lines 409-412 in the tracked changes version of the manuscript and new Figure S6D-F. The results obtained confirm the MTTbased results (Fig.6H-I).

Results:(1) Fig. 5G: The number of BK "positive" mitoplasts is surprisingly low - how does this affect the interpretation? Did the authors attempt to record mitoBK current in the "whole-mitoplast" mode? How does the mitoBK current density compare with that of the plasma membrane? Is it possible to theoretically predict the number of mitoBK channels per mitochondrion to elicit the observed effects? Can these results be correlated with the immuno-localization of mitoBK channels?

Indeed, the number of BKCa-positive mitoplasts appears low on a first view. However, as these experiments were performed in a mitoplast-attached mode, it is important to keep in mind that only a very small area of the actual mitoplast is investigated with each patch. If no channel was detected in such region, the patch was depicted as “empty”, as presented in Fig.5G, which does, however, not mean that the entire mitochondria was actually BKCa negative. Hence, the density of BKCa in the IMM might be higher than expected from our experiments. Nevertheless, already earlier results using glioblastoma cell lines – considered to be one of the cell lines mostly enriched in mitoBKCa – demonstrated a quite low density of BKCa β4 regulatory subunit in mitochondria – please see figure 2B in the following paper: 10.1371/journal.pone.0068125 – which (based on 1:1 stoichiometry of α and β subunits) also suggests that the density of the alpha subunit of BKCa might be low in this compartment.

**Author response image 3. sa4fig3:** Author response image 3: Schematic representation of mitoplast attached patch-clamp experiments.

Theoretically, density predictions of mitoBK compared to PM localized BKCa would be possible if whole-mitoplast experiments were performed, however, we are unsure what added value this information would actually burst, allowing the pharmacologic modulation of structures originally located within the mitochondrial matrix. Please also consult Author response image 3. According to the most recent models, even if there are other views on this, mitoBKCa is oriented in a way, that the C-terminus with its Ca2+ binding bowl is located within the mitochondrial matrix. Hence, to allow Ca2+ sensitivity experiments of the channel, broken up (by swelling) mitoplasts are required to make the Ca2+ binding bowl accessible for Ca2+ manipulations in the bath solution. This approach does not allow us to compare the channel density to that of the PM.

Finally, to the best of our knowledge, a combination of immunofluorescence with mitoplast patch-clamp experiments is not feasible yet, and would probably be impossible due to the low density of the mitoBKCa as well as the lack of highly sensitive and specific antibodies.

(2) There are also reports about other mitoK channels (e.g. Kv1.3, KCa3.1, KATP) playing an important role in mitochondrial function. Did the authors observe them, too? Can the authors speculate on the relative importance of the different channels? Is it known whether they are expressed organ-/tumor-specifically?

**Author response image 4. sa4fig4:** Representative single channels different to mitoBKCa detected in MDAMB-453 mitoplasts.

The reviewer is right, other K+ channels have been found in mitochondria and these also play a role in tumor cells. This is also consistent with our data (Fig.5G), where we observed other channels in the mitoplasts of BCCs as well. These all four cell lines tested. According to their conductance and our expectations from literature, these channels may e.g. include mitoIKCa, mitoSKCa, mitoKATP orothers (10.1146/annurev-biophys-092622-094853). As we focused, however, on patches containing a mitoBKCa, we did not further pharmacologically characterize these channels. Two examples of channels we found in these mitoplasts besides BKCa are presented for reviewers’ inspection (Author response image 4). As our manuscript focusses on mitoBKCa, we did not further classify these channels in smaller subgroups according to their conductance, as we feel that a differentiation between BKCa (~210 pS), and channels showing a conductance ≤150pS, or a conductance ≤100 pS is sufficient. Furthermore, this additional information would dilute our story too much making it difficult for the (non-specialist) reader to follow the red thread of the study. We added respective information in the manuscript, however. Please consult lines 365-366 in the tracked changes version of the manuscript.

Reviewer #1 is right, the observed the different K+ channels might of course be organ- or tumor-specific. For example, it has been reported that the expression of K+ channels is different in various cancer cell (lines) (https://doi.org/10.2174/13816128113199990032, 10.1016/j.pharmthera.2021.107874, 10.1038/nrc3635), a fact, which also according to our study might be exploited for pharmacological manipulation, aiming to affect proliferation/apoptosis of cancer cells. Further, a recently published single-cell and spatially resolved atlas of human breast cancer implies that the expression of different K+ channels (such as mitoIKCa, mitoSKCa, mitoKATP) might even differ between cancer- and non-cancer cells within a single tumour (https://doi.org/10.1038/s41588-021-00911-1).

**Reviewer #2 (Public Review):**
Summary:The large-conductance Ca2+ activated K+ channel (BK) has been reported to promote breast cancer progression, but it is not clear how. The present study carried out in breast cancer cell lines, concludes that BK located in mitochondria reprograms cells towards the Warburg phenotype, one of the metabolic hallmarks of cancer.Strengths:The use of a wide array of modern complementary techniques, including metabolic imaging, respirometry, metabolomics, and electrophysiology. On the whole, experiments are astute and well-designed and appear carefully done. The use of BK knock-out cells to control for the specificity of the pharmacological tools is a major strength. The manuscript is clearly written.There are many interesting original observations that may give birth to new studies.Weaknesses:The main conclusion regarding the role of a BK channel located in mitochondria appears is not sufficiently supported. Other perfectible aspects are the interpretation of co-localization experiments and the calibration of Ca2+ dyes. These points are discussed in more detail in the following paragraphs:

We thank reviewer #2 for the thorough assessment of our study.

(1) May the metabolic effects be ascribed to a BK located in mitochondria? Unfortunately not, at least with the available evidence. While it is clear these cells have a BK in mitochondria (characteristic K+ currents detected in mitoplasts) and it is also well substantiated that the metabolic effects in intact cells are explained by an intracellular BK (paxilline effects absent in the BK KO), it does not follow that both observations are linked. Given that ectopic BKDEC appeared at the surface, a confounding factor is the likely expression of BK in other intracellular locations such as ER, Golgi, endosomes, etc. To their credit, authors acknowledge this limitation several times throughout the text ("...presumably mitoBK...") but not in other important places, particularly in the title and abstract.

We thank the reviewer for this important comment and amended the title and abstract, respectively. The title of the manuscript was changed to “mitoBKCa is functionally expressed in murine and human breast cancer cells and potentially contributes to metabolic reprogramming.” Additionally, we changed appropriate passages in the text, to emphasize that mitoBKCa potentially mediates the metabolic reprogramming, but other intracellular channels could also contribute to these processes.

(2) MitoBK subcellular location. Pearson correlations of 0.6 and about zero were obtained between the locations of mitoGREEN on one side, and mRFP or RFP-GPI on the other (Figs. 1G and S1E). These are nice positive and negative controls. For BK-DECRFP however, the Pearson correlation was about 0.2. What is the Z resolution of apotome imaging? Assuming an optimum optical section of 600 nm, as obtained by a 1.4 NA objective with a confocal, that mitochondria are typically 100 nm in diameter and that BK-DECRFP appears to stain more structures than mitoGREEN, the positive correlation of 0.2 may not reflect colocalization. For instance, it could be that BK-DECRFP is not just in mitochondria but in a close underlying organelle e.g. the ER. Along the same line, why did BK-RFP also give a positive Pearson? Isn´t that unexpected? Considering that BK-DEC was found by patch clamping at the plasma membrane, the subcellular targeting of the channel is suspect. Could it be that the endogenous BK-DEC does actually reside exclusively in mitochondria (a true mitoBK), but overflows to other membranes upon overexpression? Regarding immunodetection of BK in the mitochondrial Percoll preparation (Fig. S5), the absence of NKA demonstrates the absence of plasma membrane contamination but does not inform about contamination by other intracellular membranes.

Indeed, it seems that BKCa-DEC is not an exclusive mitoBKCa, at least not upon (over-/)expression in MCF-7 cells. It is known from literature, that mitochondrial K+ channels are encoded by the nuclear genome, as no obvious gene for a K+ channel is found in the mitochondrial genome. Channel proteins are synthetized by cytosolic ribosomes and likely translocated into mitochondria via the TOM/TIM system. Although some K+ channels possess a mitochondrial targeting sequence at the N-terminus, their import is mostly far from a general mechanism, and this seems also to be true for BK channels. In the case of the K+ channel Kv1.3, an even more complex scenario is hypothesized, as the channel located in the PM could be transferred to mitochondria via mitochondria-associated membranes (MAM) structures of the ER (https://doi.org/10.3390/ijms20030734). Yet, the detailed mechanism for BK shuttling to mitochondria is not fully understood. Possibly, overflow is exactly what is happening, due to very high levels of BK-DEC expression upon transfection. However, that the channel translocates to the IMM upon transfection is not surprising and was also demonstrated for other cell models including HEK293 – see e.g. 10.1038/s41598-021-904653. Unfortunately, transfection efficiency of MCF-7 is quite low compared to HEK293 – hence, quantitative statements from mito-patches upon transfection are difficult.

In order to ensure that the mitochondrial colocalization is not a matter of poor microscope resolution, we reperformed these experiments using confocal imaging on a Zeiss LSM980 with an Airyscan 2 detector, yielding z resolutions of ~ 450 nm. These experiments confirmed the increased colocalization of BKCa-DEC with mitochondria compared to BKCa lacking the DEC exon. Furthermore, this imaging at higher resolution demonstrated, that, unfortunately, colocalization might not be the best analysis, as especially fragmented mitochondria showed a clear MitoGREEN stained matrix, surrounded by red fluorescence derived from BKCaDECRFP present in the IMM (revised Fig. 1G).

To validate the results derived from immunoblotting, we additionally stained the membranes for TMX1, a marker for the ER membrane. This analysis confirmed the high purity of the mitochondrial isolation without ER-membrane contamination after percoll purification, and hence validated the presence of BKCa in the mitochondrial membrane (revised Fig. S5D). The additional information can be found in lines 156-159 in the tracked changes version of the manuscript.

(3) Calibration of fluorescent probes. The conclusion that BK blockers or BK expression affects resting Ca2+ levels should be better supported. Fluorescent sensors and dyes provide signals or ratios that need to be calibrated if comparisons between different cell types or experimental conditions are to be made. This is implicitly acknowledged here when monitoring ER Ca2+, with an elaborate protocol to deplete the organelle in order to achieve a reading at zero Ca2+.

We thank the reviewer for the important comment. Please note that at no point in the manuscript we aim to compare different cell lines concerning their intracellular Ca2+ concentration, but we only compare the same cell lines after the different treatments, as we are aware of this limitation of fluorescent probes. However, to validate the differences in intracellular Ca2+ concentrations, we calibrated the signals derived from Fura-2 and 4mtD3cpV using ionomycin in combination with cellular Ca2+ depletion/ saturation. The newly added data can be found in the text, lines 185-190, 192-195, 199-200, and 228-230 in the tracked changes version of the manuscript, as well as new Figure S2F – Figure S2J and new Figure S2R – Figure S2V

Line 203. "...solely by the expression of BKCa-DECRFP in MCF-7 cells". Granted, the effect of BKCa-DECRFP on the basal FRET ratio appears stronger than that of BK-RFP, but it appears that the latter had some effect. Please provide the statistics of the latter against the control group (after calibration, see above).

**Author response image 5. sa4fig5:** Dot blot for data shown in Figure 2I.

The reviewer is right, it seems that BKCaRFP may also affect [Ca2+]mito. However, the effect is not significant and shows a p-value of p>0.999 using Kruskal-Wallistest followed by Dunn’s multiple comparison test, due to the non-normally distributed nature of the data. p=0.0002 for ctrl vs. BKCa-DECRFP and 0.0022 for BKCaRFP vs. BKCa-DECRFP, however. We added a scatter dot-blot of the respective data as Author response image 5 for reviewer’s inspection. Additionally, first, even using a more stringent statistical test by only comparing ctrl vs BKCaRFP using Mann-Whitney test, the results are not significant, as the p-value was determined at 0.4467, and second, we performed the requested Ca2+calibration using ionomycin under these conditions, which confirmed the difference between ctrl cells and BKCa-DECRFP expressing cells, but not BKCaRFP expressing ones. Please see Figure S2V.

**Reviewer #3 (Public Review):**
The original research article, titled "mitoBKCa is functionally expressed in murine and human breast cancer cells and promotes metabolic reprogramming" by Bischof et al, has demonstrated the underlying molecular mechanisms of alterations in the function of Ca2+ activated K+ channel of large conductance (BKCa) in the development and progression of breast cancer. The authors also proposed that targeting mitoBKCa in combination with established anti-cancer approaches, could be considered as a novel treatment strategy in breast cancer treatment.The paper is clearly written, and the reported results are interesting.Strengths:Rigorous biophysical experimental proof in support of the hypothesis.Weaknesses:A combinatorial synergistic study is missing.

We thank reviewer #3 for the positive summary of our study. Indeed, we propose that targeting of mitoBKCa in combination with established anti-cancer drugs may represent a novel anti-cancer treatment strategy. Unfortunately, we feel that the manuscript is very condensed already, and that adding respective required experiments and data to support this hypothesis will make the flow of the manuscript more complex or even incomprehensible. As no attempts linking mitoBKCa activity with anti-cancer therapies have been made so far, we removed the respective information from the abstract and only discuss this aspect.

**Recommendations for the authors:**

**Reviewer #1 (Recommendations For The Authors):**
(1) Statistics: Legends have to contain information about the number of biological replicates (N) and cells analysed (n). Statistics must be calculated with the averages of the replicates.

**Author response image 6. sa4fig6:** Representative single cell responses of Fura-2 loaded MMTV-PyMT WT cells.

We thank the reviewer for the comment and added the missing details to all figure legends.

We feel that using each cell represents exactly the power of high-resolution live-cell imaging, as there is no better biological replicate than a single separated cell, which is observed by fluorescence microscopy. This analysis is also able to visualize cell-to-cell differences in the microscopy area, similarly to patch-clamp experiments, where each single cell or mitoplast patched is used as a single replicate. Please find a representative dataset derived from fluorescence microscopy of different responses of neighboring single cells in Author response image 6.

(2) Fig. 1G: This is a poor resolution figure, mostly because of its far too small size; in its current form it bears very little information.

We agree with reviewer #1 and reperformed the imaging experiments using high resolution confocal imaging and exchanged the respective images. We feel that this increased the quality of the images significantly. Unfortunately, we were not able to increase the size of the images in the main figure, hence, we added magnifications of the respective images as new Figure S1I.

(3) Fig. 1H: What do the dotted grey lines and the labels stand for?

We believe Reviewer #1 is probably referring to Figure 1G. As indicated in the figure panel and in the text, the grey dotted lines and labels indicate the colocalization scores of mtRFP and RFP-GPI with MitoGREEN, respectively. These data are also shown in Figure S1H, including error bars and statistics. We added additional information in the text to make the meaning of the lines clearer to the reader. Please consult lines 149 – 150 in the tracked changes version of the manuscript.

**Reviewer #2 (Recommendations For The Authors):**
(1) May the metabolic effects be ascribed to a BK located in mitochondria? Short of a way to tackle BK function and metabolism specifically in mitochondria, the conclusion may best be toned down to "intracellular BK". For the time being the term "mitoBK" appears too ambitious.

We fell that you are right and that our previous overstatement requires adaptation as a clear (100%) attribution of the observed metabolic effects solely to mitoBKCa is not definitely possible. We have therefore amended all relevant passages in the entire MS accordingly.

(2) MitoBK subcellular location. Please address the points raised in the Public Review.

As stated above we addressed the point raised in the public review accordingly (please consult new Figure S1I and revised Figure 1G).

(3) Calibration of fluorescent probes. Please provide calibrations for cytosolic and mitochondrial Ca2+, for example, the standard high Ca2+/ionophore/metabolic inhibition treatment to reach saturation followed by Ca2+ chelation to obtain zero Ca2+.

We thank the reviewer for the comment. As you can see from our response to the public review, we performed the respective experiments, and datasets were added in the manuscript.

(4) Line 203. "...solely by the expression of BKCa-DECRFP in MCF-7 cells". Granted, the effect of BKCa-DECRFP on the basal FRET ratio appears stronger than that of BK-RFP, but it appears that the latter had some effect. Please provide the statistics of the latter against the control group (after calibration, see above).

Please consult our response to the (same) comment in the public review.

(5) Line 228. The statement "Similar results were obtained in MDA-MB-453 cells" is confusing. As shown in Fig.3, pax reduced ECAR and OCR in MMTV-PyMT WT cells. As ibtx was without effect, it is suggested that intracellular BK support metabolism. However, the effect of pax on MDA cells was the opposite. Doesn´t this divergence speak against a universal role of intracellular BKs in promoting metabolism in BCCs? A similar point may be made regarding metabolomics, which showed no effects of pax on lactate and pyruvate in MMTV-PyMT WT cells but stimulation in MDA cells. Perhaps the word "promotes" in the title of the figure should be replaced by something more neutral like "affects" or "alters", as used elsewhere,

We thank the reviewer for pointing out the overstatement regarding intracellular BK functions and changed the title of the figure as suggested.

With regard to the experiments mentioned, we would like to point out the following aspects:

First, the cell lines used strongly differ in their metabolic settings under basal conditions. While both, MMTV-PyMT and MDA-MB-453 cells seem to show similar basal ECAR levels (if BKCa was present), their OCR seems to differ strongly. MMTV-PyMT cells seem to show a basal OCR which is almost at the maximum already, while MDA-MB-453 cells possess a tremendous capacity in their OCR, as observed upon mitochondrial uncoupling using FCCP. Of note, both, ECAR and OCR are indirect metabolic measures. On the one hand, ECAR measures extracellular acidification, which is accomplished by H+ along with lactate secretion. However, lactate secretion is not the only process leading to extracellular acidification, and ECAR may hence measure a variety of H+ releasing processes, including processes of vesicle secretion. On the other hand, OCR is not directly linked to ATP production, as mitochondrial complex IV is consuming O2, ATP, however, is produced by mitochondrial complex V. This becomes even more evident when having a look on OCRs after FCCP treatment – under these conditions, the H+ gradient is destroyed and ATP synthase activity is reduced, OCR, however, increases to the maximum due to increased supply of mitochondrial complex IV with H+.

Second, please note that the LC-MS-based metabolomics derive from a static single time point and not from an over-time “live” read-outs. Moreover, underlying dynamics of the parameters measured can not be assessed. Hence, as an example, increasing levels of pyruvate can e.g. indicate faster generation, or slower subsequent degradation/ metabolization. A clear in-depth statement about what is happening under basal and BKCa inhibitor treated conditions is hence not possible. The only conclusion possible to draw from these experiments is that paxilline treatment differentially affects metabolic pathways in these cells.

Based on these limitations of both methods, we decided to perform our in-depth fluorescence microscopy-based analysis, which provided strong evidence for intracellular BKCa channels on mitochondrial ATP production. Despite opposing effects of BKCa inhibition on OCR in MMTV-PyMT WT and MDA-MB-453 cells, mitochondrial ATP production was reduced, if BKCa-DECRFP was expressed/ intracellular BKCa was functional.

In line with these findings, mitoBKCa was recently described as an uncoupling protein, which could furthermore explain the differential effects of intracellular BKCa inhibition on OCR. https://doi.org/10.1038/s41598-021-90465-3

Minor(6) Fig. 1C. Average fluorescence intensity in 6 experiments was about 20% higher in BK-KO cells relative to WT. Such a small difference is significant but should not be evident to the eye. The pictures selected for illustration appear to show a much larger difference and therefore may not be representative. If this is the case, please omit them. The same goes for the other representative pictures.

**Author response image 7. sa4fig7:** Representative images at different brightnesses.

Please note, that the analysis of the images was done in an unbiased way using a Fiji macro. After analysis, we chose representative images, which were closest to the average.

Furthermore, we must kindly disagree with the reviewer as changes of 20% in fluorescence intensity are indeed evident to the eye (consult Author response image 7). This panels show the same image at different brightness levels with intensity differences of 20%. Hence, we feel, that all the images the reviewer was referring are representative for the values given.

(7) Line 130. The definition of "recent" is of course relative, but 10 years?

We are very glad that you have discovered this “inconsistency", and reworded the respective phrase accordingly.

(8) Line 327. "conductivity" is the property of a medium, "conductance" is the property of a component, such as a channel.

We thank the reviewer for the important comment. We revised the text accordingly.

(9) Various figures. FRET sensor data are expressed as Ratio(FRET/CFP). This is unusual, typically it should be FRET ratio (YFP/CFP), FRET ratio(mTFP/Venus), etc. Please note that the FRET partners differ between sensors.

We acknowledge the comment of the reviewer. It is correct that fluorescent proteins vary widely between the sensors (used). Please note, however, the following: The emission measured from these sensors actually represents FRET, as CFP but not YFP is directly excited. Hence, emission is FRET, not the “intrinsic” fluorescence of the YFP. This is getting more and more important to differentiate, as there are probes existing, which can also be “alternately” excited, i.e. CFP and YFP separately, which will then yield the YFP/CFP ratio (https://doi.org/10.1021/acssensors.8b01599). In case of only CFP excitation, we feel, that the term FRET/CFP is preferable over other labelings such as YFP/CFP.

(10) BK-DEC makes BCCs cells less oxidative. However, BK-DEC was first described in cardiomyocytes, which are among the most oxidative cell types. It would be useful if authors could address this apparent contradiction in the Discussion Section.

That is an exciting point that we addressed as follows in the revised MS:

First, it is important to mention that cardiac myocytes do not show a metabolic Warburg setting and are – under physiologic conditions – maintained in a high O2 environment.

Second, a recent study from our group addressed the question about the role of mitoBKCa in primary cardiac myocytes. Indeed, mitoBKCa was functionally expressed in these cells. Interestingly, under physiologic conditions, the channel did not alter (multiple) cell behaviours nor overall cardiac physiology in a mouse model. However, upon induction of ischemia/ reperfusion injury, a lack of BK increased cardiac susceptibility to cell death resulting in increased infarction size (https://doi.org/10.1161/CIRCULATIONAHA.117.028723). Hence, also in this cell model, BKCa only played a role under oxygen limited conditions/ conditions where mitochondria were not properly functioning. Thus, the results derived from cardiac myocytes support our recent findings in BCCs, as BKCa mediates BCC resistance to hypoxic stress/ makes BCCs more independent from oxidative metabolism.

Parts of this discussion were included in the revised MS. Please consult lines 490-500 in the tracked changes version of the manuscript.

**Reviewer #3 (Recommendations For The Authors):**
(1) The study is very well designed and most of the computational analyses were done rigorously.

We highly appreciate the positive feedback by reviewer #3.

(2) The authors should discuss the expression of BKCa in different subsets of breast cancer. Authors may also debate on the level of steroid receptors and BKCa expressions.

We thank reviewer #3 for the important suggestion and added the requested information in the discussion, lines 445-447 and 450-454 in the tracked changes version of the manuscript.

(3) In the discussion section, the authors mentioned that the MCF7 cell is the best model to study this hypothesis. Does it imply that triple-negative breast cancer cell lines express lower levels of BKCa? The authors should discuss this.

We thank the reviewer for the interesting comment; we would like to point out that the ERα-positive MCF-7 cell line was used to study experimental overexpression of BKCa at an otherwise low baseline level. This does not imply that BKCa is expressed at lower levels in TNBC cell lines; in fact a recent study showed the opposite, i.e. overexpression of BKCa in TNBC patients (10.1186/s12885-020-07071-1). Consistent with our work, the authors conclude that the channel could even be a new strategy for development of a targeted therapy in TNBC. We also added this information in the discussion, lines 450-454 in the tracked changes version of the manuscript.

(4) The authors propose that combinatorial targeting of mitoBKCa along with known breast cancer chemotherapeutics can open a new horizon in breast cancer treatment. However, the authors did not perform any experiment to show the synergistic effect as mentioned.

As already stated in the public reviews, we feel that the manuscript is very condensed already, and that adding the respective experiments and data will make the flow of the study even more complex. For the moment, we removed all information and statements linking mitoBKCa with anti-cancer treatment strategies from the abstract and only discuss this aspect. We hope that the reviewer agrees with us that an extensive analysis of the functional mitoBKCa status in the context of established breast cancer therapies must be addressed by (our) future studies.

Minor Comments:There are several typos and grammatical errors that need further attention and rephrasing.

We thank the reviewer for the comment and revised the text accordingly.